# *Met* and *Cxcr4* cooperate to protect skeletal muscle stem cells against inflammation-induced damage during regeneration

Ines Lahmann[1,2], Joscha Griger[2†], Jie-Shin Chen[2‡], Yao Zhang[2], Markus Schuelke[3], Carmen Birchmeier[1,2]*

[1]Neurowissenschaftliches Forschungzentrum, NeuroCure Cluster of Excellence, Charité–Universitätsmedizin Berlin, Corporate Member of Freie Universität Berlin and Humboldt-Universität zu Berlin, Berlin, Germany; [2]Developmental Biology/Signal Transduction Group, Max Delbrueck Center for Molecular Medicine (MDC) in the Helmholtz Society, Berlin, Germany; [3]Department of Neuropediatrics, Charité–Universitätsmedizin Berlin, Corporate Member of Freie Universität Berlin and Humboldt-Universität zu Berlin, Berlin, Germany

*For correspondence:
cbirch@mdc-berlin.de

Present address: †Center for Translational Cancer, Technische Universität München, Munich, Germany; ‡CVMD, IMED Biotech Unit, AstraZeneca Gothenburg, Mölndal, Sweden

**Abstract** Acute skeletal muscle injury is followed by an inflammatory response, removal of damaged tissue, and the generation of new muscle fibers by resident muscle stem cells, a process well characterized in murine injury models. Inflammatory cells are needed to remove the debris at the site of injury and provide signals that are beneficial for repair. However, they also release chemokines, reactive oxygen species, as well as enzymes for clearance of damaged cells and fibers, which muscle stem cells have to withstand in order to regenerate the muscle. We show here that MET and CXCR4 cooperate to protect muscle stem cells against the adverse environment encountered during muscle repair. This powerful cyto-protective role was revealed by the genetic ablation of Met and Cxcr4 in muscle stem cells of mice, which resulted in severe apoptosis during early stages of regeneration. TNFα neutralizing antibodies rescued the apoptosis, indicating that TNFα provides crucial cell-death signals during muscle repair that are counteracted by MET and CXCR4. We conclude that muscle stem cells require MET and CXCR4 to protect them against the harsh inflammatory environment encountered in an acute muscle injury.

## Introduction

Muscle injury through trauma is common and can be repaired by muscle regeneration (*Järvinen et al., 2005*; *Tidball, 2005*; *Tidball, 2017*). Stem cells reside in the muscle tissue and provide the cellular source for the regeneration process (*Chargé and Rudnicki, 2004*; *Relaix and Zammit, 2012*). Muscle stem cells are characterized by the expression of PAX7 and their location in the stem cell niche between the basal lamina and plasma membrane of the muscle fiber (*Mauro, 1961*; *Seale et al., 2000*). Muscle stem cells are quiescent in the adult, but can be re-activated upon injury. On one hand, activated muscle stem cells proliferate and generate differentiating cells to repair the muscle, and on the other they can self-renew to repopulate the stem cell niche (*Chargé and Rudnicki, 2004*; *Relaix and Zammit, 2012*). A complex interplay between muscle stem cells and their environment occurs during muscle repair. Inflammatory cells and the cytokines they produce provide important cues for muscle stem cells and regulate their activation, proliferation, and differentiation. Therefore, communication between muscle stem cells and the immune system needs to be tightly regulated. Failure of

adequate communication results in incomplete regeneration as well as sustained or chronic inflammation that ultimately damages the muscle (*Chazaud et al., 2009*; *Saclier et al., 2013b*; *Londhe and Guttridge, 2015*; *Tidball, 2017*).

Shortly after an acute muscle injury, resident macrophages are activated and large numbers of macrophages and neutrophils are recruited to the injured tissue. This accumulation of immune cells is a prerequisite for the removal of damaged fibers (*Tidball, 2005*). The immune cells amplify the inflammatory response and create a milieu that is rich in inflammatory cytokines, reactive oxygen species, proteases, and membrane-damaging agents (*Butterfield et al., 2006*; *Mann et al., 2011*; *Le Moal et al., 2017*). This produces a noxious environment that muscle stem cells and regenerating fibers must withstand in order to properly rebuild functional muscle tissue. How muscle stem cells are protected from these noxious cues has not yet been elucidated.

We reasoned that investigating the direct role of cytokines on muscle stem cells and during muscle repair after acute injury will help to define factors that could be beneficial in a therapeutic setting. We used cardiotoxin injection as our muscle injury model that resulted in widespread necrosis of muscle fibers, massive infiltration by neutrophils and macrophages followed by myogenic regeneration. In such a setting, extensive proliferation of muscle stem cells occurs, amplifying their numbers and providing the cellular material for new myofibers (*Hardy et al., 2016*). Nevertheless, a substantial number of stem cells are lost during the acute inflammatory response (*Hardy et al., 2016*).

We show here that endogenous cytokines enable muscle stem cells to survive in the noxious environment encountered after injury. We used mouse genetics to demonstrate that MET/HGF and CXCR4/CXCL12 signals cooperate to protect muscle stem cells during early stages of regeneration. We identify TNFα as a factor that damages the stem cells in this setting. Together, our study shows that inflammatory factors have dual effects, damaging (TNFα) and protecting (HGF and CXCL12) muscle stem cells during acute injury and regeneration.

## Results
### *Met* is required for normal muscle regeneration

To identify factors that directly regulate muscle stem cell behavior in vivo, we systematically assessed chemokine transcripts in regenerating muscle using published data sources (*Hirata et al., 2003*; *Xiao et al., 2011*; *Bobadilla et al., 2014*) and verified their expression using qPCR. A multitude of chemokines are rapidly and strongly induced after injury. In murine tibialis anterior muscle tissue, *Tnf* and *Hgf* transcripts were induced 10–500-fold with a time course that peaked 2–3 days after injury (*Figure 1A and B*, *Figure 1—figure supplement 1*, and *Supplementary file 1*). TNFα is known to orchestrate the inflammatory response and to participate in the communication between immune cells (*Saclier et al., 2013a*; *Turner et al., 2014*), and HGF is a proliferation and motility factor that can act as protective factor in tissue injury (*Birchmeier et al., 2003*; *Nakamura and Mizuno, 2010*). *Hgf* transcripts were produced at low levels by quiescent and activated muscle stem cells, demonstrating that other cell types but muscle stem cells produce *Hgf* in the regenerating muscle (*Figure 1C* and *Supplementary file 1*). This is in accord with previous data on *Hgf* expression obtained by microarray analysis (*Liu et al., 2013*; *Latroche et al., 2017*; see also *Figure 1—figure supplement 1*). The HGF receptor MET is expressed in adult muscle stem cells (*Cornelison and Wold, 1997*), and, in contrast to quiescent muscle stem cells, *Met* transcripts were upregulated when the cells were activated (*Figure 1D*).

To identify the role of HGF/MET during muscle repair, we introduced a loss-of-function mutation in *Met* in muscle stem cells using a constitutive *Pax7iresCre* allele (*Pax7iresCre;Metflox/flox* mice, named hereafter co*Met;* the genotype of the corresponding control mice used was *Pax7iresCre;Met+/+*). *Met* is known to control migration of myogenic progenitors during development (*Bladt et al., 1995*). The conditional mutation did not affect muscle progenitor migration because *Pax7* (and hence *Pax7iresCre*) starts only to be expressed in progenitors that have already reached their targets (*Relaix et al., 2004*). Therefore, the *Met* mutation in myogenic progenitors is introduced after migration is completed, from there on persisting throughout fetal and postnatal development. In the undamaged muscle, neither fiber diameter nor muscle stem cell numbers were changed in co*Met* mutant compared to control mice (*Figures 2 and 3*).

Upon injury of the *tibialis anterior* muscle using cardiotoxin, co*Met* mutant muscle stem cells were able to regenerate muscle fibers. However, at 7 days post injury (dpi) the diameter of the newly

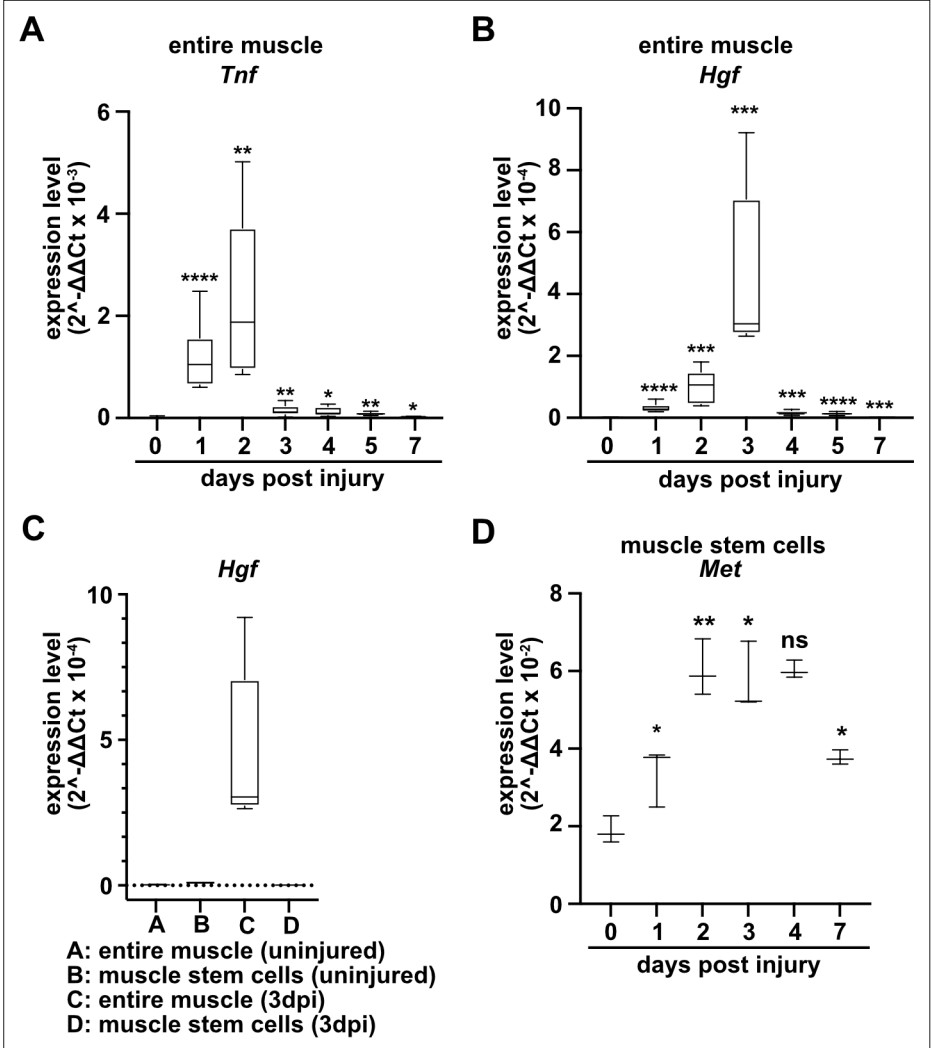

**Figure 1.** Expression of *Tnf*, *Hgf*, and *Met* during muscle regeneration. (**A, B**) Expression dynamics of *Tnf* (**A**) and *Hgf* (**B**) in uninjured and regenerating muscle tissue determined by qPCR. (**C**) Expression dynamics of *Hgf* in quiescent and activated muscle stem cells and in muscle tissue during muscle regeneration determined by qPCR. (**D**) Expression levels of *Met* in quiescent and activated muscle stem cells determined by qPCR. Boxplots represent interquartile range, and whiskers show min-to-max range. β-Actin expression was used for normalization in (**A–D**).

The online version of this article includes the following source data and figure supplement(s) for figure 1:

**Source data 1.** Quantification of *Tnf*, *Hgf* and *Met* expression represented in the diagrams shown in A-D.

**Figure supplement 1.** Expression levels of *Hgf* in quiescent (freshly isolated) and proliferating muscle stem cells at various time points defined by microarray analysis (*Liu et al., 2013*; *Latroche et al., 2017*).

regenerated fibers was smaller in co*Met* mutants than in control animals, but diameters largely equalized between control and co*Met* mutants and were no longer significantly different at 20 dpi (*Figure 2A–D*, quantified in E). Moreover, the number of PAX7+ stem cells in the regenerated muscle of co*Met* mice was reduced by 68% at 7 dpi compared to control mice, and also this difference became less pronounced at 20 dpi (47% reduction in co*Met* mice; *Figure 3A–F*). Similar deficits were observed when *Met* was mutated in adult muscle stem cells using the tamoxifen-inducible *Pax7^iresCreERT2* allele (*Pax7^iresCreERT2Gaka/+;Met^flox/flox* mice treated with tamoxifen, named hereafter Tx^Gaka*Met* as controls, *Pax7^iresCreERT2Gaka/+;Met^+/+* mice treated with tamoxifen were used). Thus, the diameter of new fibers was smaller at 7 dpi in Tx^Gaka*Met* compared to control animals at 7 dpi, but at 20 dpi the difference in fiber diameters was no longer significant (*Figure 2F-J*). Moreover, the number of PAX7+ stem cells in the regenerated muscle of Tx^Gaka*Met* mice was reduced by 73% at 7 dpi compared to control, and also

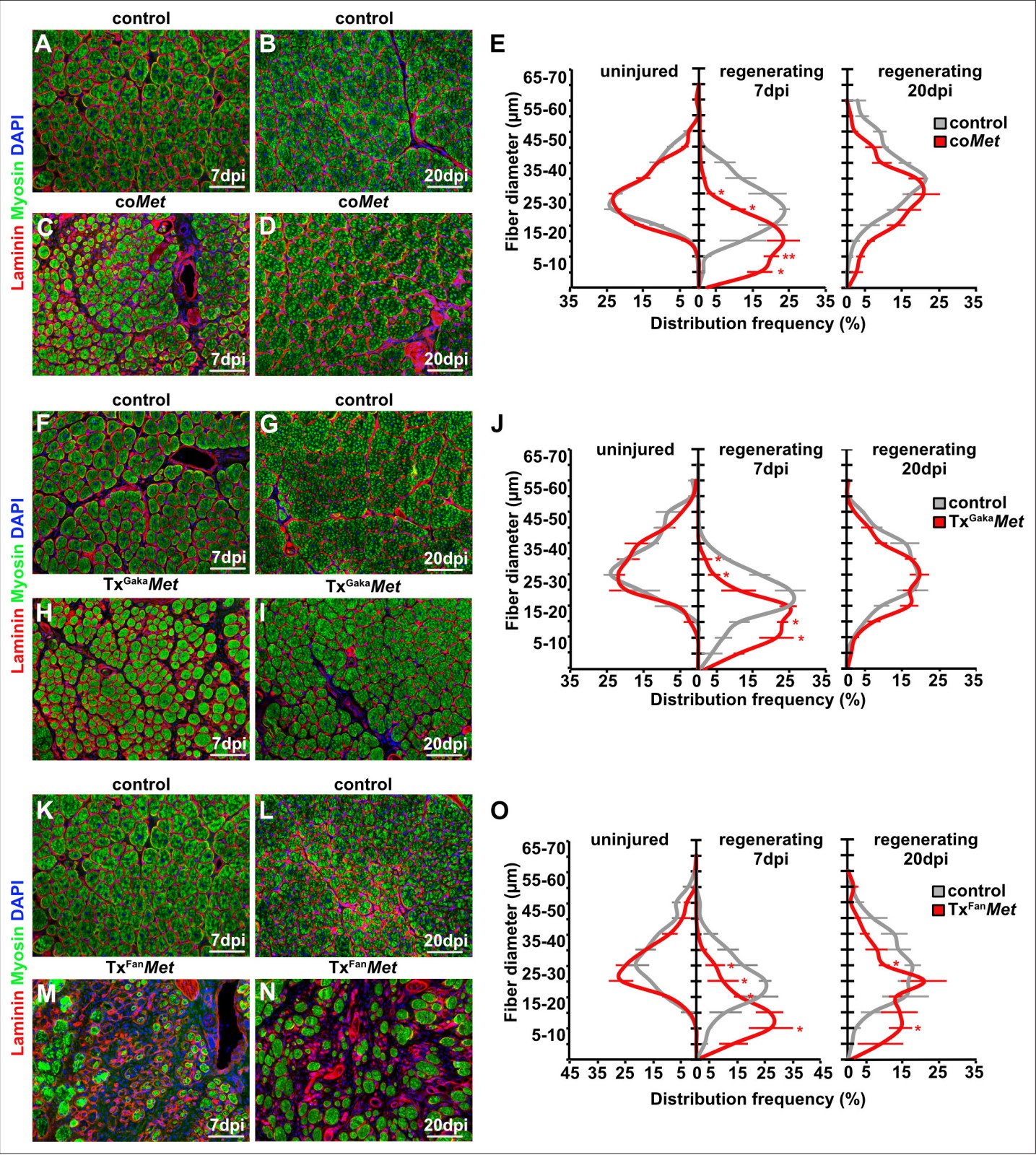

**Figure 2.** Mutation of *Met* impairs muscle regeneration. (**A–D**) Immunohistological analysis of regenerating (7 days post injury [dpi] and 20 dpi) muscle of control and co*Met* mutants using antibodies against laminin (red) and sarcomeric myosin (green). DAPI was used as a counterstain. (**E**) Distribution of Feret fiber diameters in uninjured and regenerating muscle (7 dpi and 20 dpi) of control mice and co*Met* mutants. (**F–I**) Immunohistological analysis of regenerating muscle of control and Tx^Gaka*Met* mice using antibodies against laminin (red) and sarcomeric myosin (green). DAPI was used

*Figure 2 continued on next page*

_Figure 2 continued_

as a counterstain. (**J**) Distribution of Feret fiber diameters in uninjured and regenerating muscle (7 dpi and 20 dpi) of control and Tx$^{Gaka}$_Met_ mice. (**K–N**) Immunohistological analysis of regenerating (7 dpi and 20 dpi) muscle of control and Tx$^{Fan}$_Met_ mice using antibodies against laminin (red) and sarcomeric myosin (green). DAPI was used as a counterstain. (**O**) Distribution of Feret fiber diameters in uninjured and regenerating (7 dpi and 20 dpi) muscle of control and Tx$^{Fan}$_Met_ mice. Scale bars, 100 μm. In (**A–E**) control: _Pax7$^{iresCre/+}$;Met$^{+/+}$_; co_Met_: _Pax7$^{iresCre/+}$;Met$^{flox/flox}$_. In (**F–J**) control: _Pax7$^{iresCreERT2Gaka/+}$;Met$^{+/+}$_; Tx$^{Gaka}$_Met_: _Pax7$^{iresCreERT2Gaka/+}$;Met$^{flox/flox}$_; In (**K–O**) control: _Pax7$^{CreERT2Fan/+}$;Met$^{+/+}$_; Tx$^{Fan}$_Met_: _Pax7$^{CreERT2Fan/+}$;Met$^{flox/flox}$_. Animals in (**F–O**) were treated with tamoxifen.

The online version of this article includes the following source data for figure 2:

**Source data 1.** Quantification of fiber diameters represented in the diagrams shown in E, J and O.

this difference was less pronounced at 20 dpi (44% reduction in Tx$^{Gaka}$_Met_ mice) (**_Figure 3G–L_**). In summary, our data indicate that loss of _Met_ in muscle stem cells results in a mild regeneration deficit. This is accompanied by a reduction of muscle stem cell numbers during early stages of regeneration, which is partly compensated for during late stages. Increased proliferation of the remaining stem cell pool might account for this (see below for a more detailed description of the mechanisms). A previous report had indicated that ablation of _Met_ using a distinct tamoxifen-inducible _Pax7$^{CreERT2}$_ allele (_Pax7$^{CreERT2Fan}$_) resulted in a much more severe muscle regeneration deficit (**_Webster and Fan, 2013_**). We used this _Cre_ allele to mutate _Met_ (_Pax7$^{CreERT2Fan/+}$;Met$^{flox/flox}$_ mice treated with tamoxifen, named hereafter Tx$^{Fan}$_Met_ animals; as controls, _Pax7$^{CreERT2Fan/+}$;Met$^{+/+}$_ mice treated with tamoxifen were used), and also detected a very severe muscle regeneration deficit at 7 dpi and 20 dpi compared to control animals at these stages of regeneration (**_Figure 2K–O_**). In particular, extracellular matrix remnants from injured skeletal muscle fibers (i.e., ghost fibers) were abundant at 7 dpi and 20 dpi. Notably, even in the uninjured muscle a 50 % reduction in the number of PAX7+ cells was observed in the Tx$^{Fan}$_Met_ animals compared to controls. This became more pronounced after injury when a 94 and 65% reduction in stem cell numbers was present at 7 dpi and 20 dpi, respectively, compared to the control animals at these stages of regeneration (**_Figure 3M–R_**). Different recombination efficacies did not account for these differences in phenotypes observed in co_Met_ and Tx$^{Gaka}$_Met_ animals on one side, and Tx$^{Fan}$_Met_ animals on the other side (**_Figure 3—figure supplement 1A-D_**). We conclude that the muscle stem cell and regeneration deficits present in Tx$^{Fan}$_Met_ mutants are apparently not only due to the _Met_ ablation. It should be noted that in the _Pax7$^{CreERT2Fan}$;_ allele, the _Pax7_ coding sequence is disrupted by _Cre_, whereas the _Pax7$^{iresCreERT2Gaka}$_ and _Pax7$^{iresCre}$_ alleles do not interfere with the _Pax7_ coding sequence (**_Keller et al., 2004_**; **_Lepper et al., 2009_**; **_Murphy et al., 2011_**; see also **_Figure 3—figure supplement 1E_** for a cartoon of the different _Cre_ alleles used). PAX7 levels are known to affect muscle stem cell behavior and their ability to regenerate the muscle (**_von Maltzahn et al., 2013_**; **_Mademtzoglou et al., 2018_**). Thus, the absence of one functional _Pax7_ allele might contribute to the exacerbated muscle stem cell and regeneration phenotypes observed in Tx$^{Fan}$_Met_ animals.

## MET and CXCR4 signaling cooperates during muscle regeneration

The CXCR4 receptor is expressed in developing and adult muscle stem cells and mediates CXCL12 signals that stimulate their proliferation and migration (**_Vasyutina et al., 2005_**; **_Odemis et al., 2007_**; **_Griffin et al., 2010_**). _Cxcl12_ is expressed by various cell types of the immune system. qPCR demonstrated that muscle tissue and PAX7+ cells expressed _Cxcl12_ transcripts in both uninjured and regenerating muscle, and confirmed that _Cxcr4_ transcripts were present in PAX7+ cells (**_Figure 4A–C_**, **_Figure 4—figure supplement 1_**, and **_Supplementary file 1_**). _Cxcr4_ and _Met_ are known to cooperate during muscle development (**_Vasyutina et al., 2005_**). We therefore tested whether this cooperativity was also observed in adult muscle stem cells and whether it would have an impact on muscle repair using _Cxcr4_ and _Met_ double mutant mice (_Pax7$^{iresCreERT2Gaka/+}$;Cxcr4$^{flox/flox}$;Met$^{flox/flox}$_ mice treated with tamoxifen, hereafter called Tx$^{Gaka}$_Cxcr4;Met_ animals; _Pax7$^{iresCreERT2Gaka/+}$;Cxcr4$^{+/+}$;Met$^{+/+}$_ treated with tamoxifen served as controls). Mutations of _Cxcr4_ and _Met_ in muscle stem cells did not obviously affect muscle formation or muscle stem cell numbers (**_Figure 5_**, **_Figure 5—figure supplement 1_**). However, Tx$^{Gaka}$_Cxcr4;Met_ double mutant mice at 7 dpi displayed a very severe regeneration deficit compared to control mice at 7 dpi. In particular, formation of myofibers was strongly impaired (**_Figure 5A–E_**). Further, the number of muscle stem cells detected at 7 dpi was decreased by 93 % as compared to control mice at 7 dpi (**_Figure 5F–J_**). The severe regeneration deficit was accompanied by widespread fibrosis, persisting macrophages,

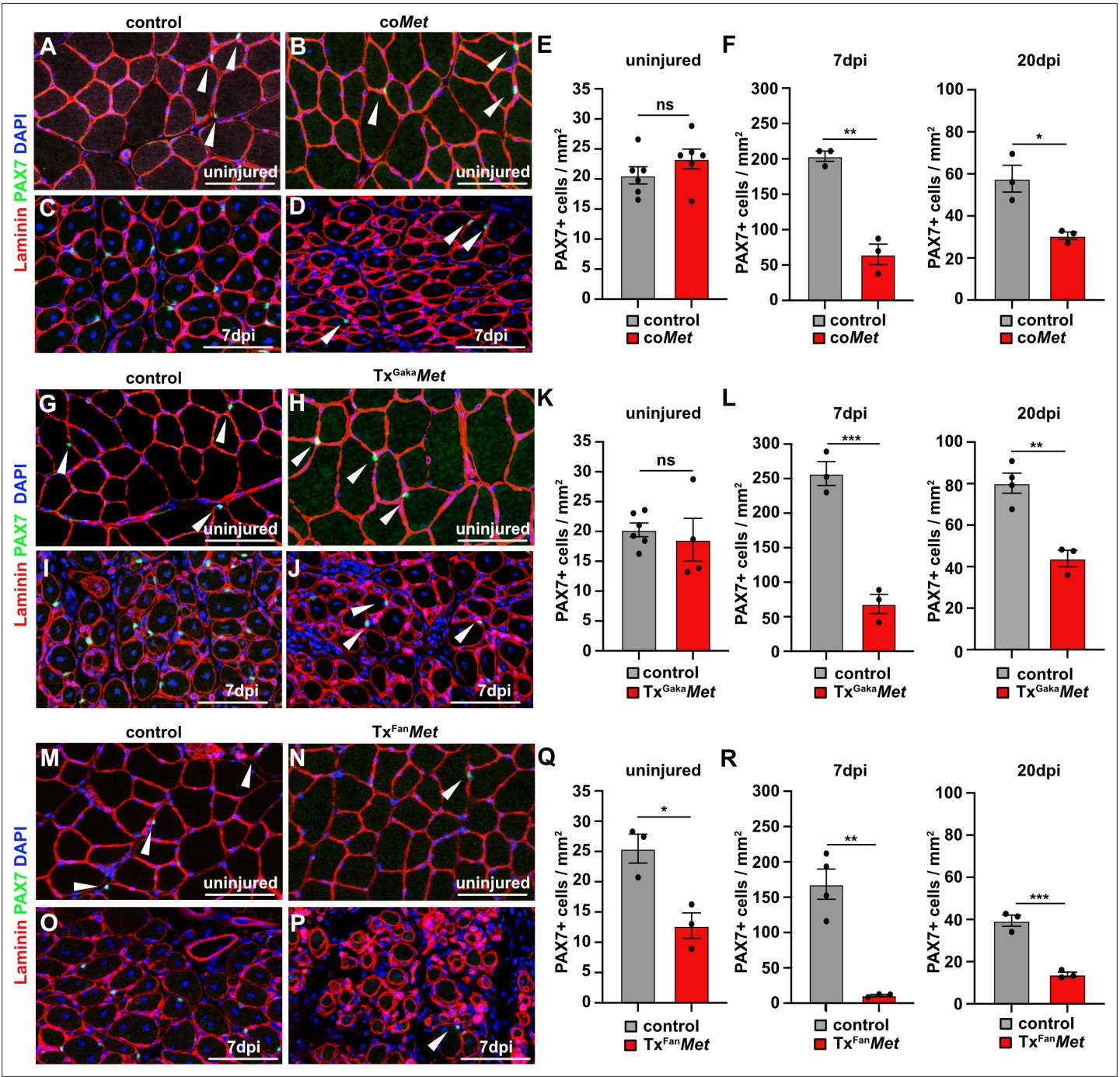

**Figure 3.** Mutation of *Met* reduces the muscle stem cell pool during regeneration. (**A–D**) Immunohistological analysis of uninjured and regenerating (7 days post injury [dpi]) muscle of control and co*Met* mice using antibodies against laminin (red) and PAX7 (green). DAPI was used as a counterstain. (**E, F**) Quantification of PAX7+ cells in uninjured and regenerating muscle from control and co*Met* mice. (**G–J**) Immunohistological analysis of uninjured and regenerating (7 dpi) muscle from control and Tx^Gaka*Met* mice using antibodies against laminin (red) and Pax7 (green). DAPI was used as a counterstain. (**K, L**) Quantification of PAX7+ cells in uninjured and regenerating muscle of control and Tx^Gaka*Met* mice. (**M–P**) Immunohistological analysis of uninjured and regenerating (7 dpi) muscle from control and Tx^Fan*Met* mice using antibodies against laminin (red) and Pax7 (green). DAPI was used as a counterstain. (**Q, R**) Quantification of PAX7+ cells in uninjured and regenerating (7 dpi) muscle from control and Tx^Fan*Met* mice. Arrowheads point to PAX7+ cells. Scale bars 100 µm. In (**A–F**) control: *Pax7^iresCre/+;Met^+/+*; co*Met: Pax7^iresCre/+;Met^flox/flox*. In (**G–L**) control: *Pax7^iresCreERT2Gaka/+;Met^+/+*; Tx^Gaka*Met: Pax7^iresCreERT2Gaka/+;Met^flox/flox*. In (**M–R**) control: *Pax7^CreERT2Fan/+;Met^+/+*; Tx^Fan*Met: Pax7^CreERT2Fan/+;Met^flox/flox*. Animals in (**G–R**) were treated with tamoxifen.

The online version of this article includes the following source data and figure supplement(s) for figure 3:

**Source data 1.** Quantification of PAX7+ cells represented in the diagrams shown in E, F, K, L, Q and R (*Figure 3*).

**Figure supplement 1.** Recombination efficiency and schematic drawing of the different Pax7Cre alleles.

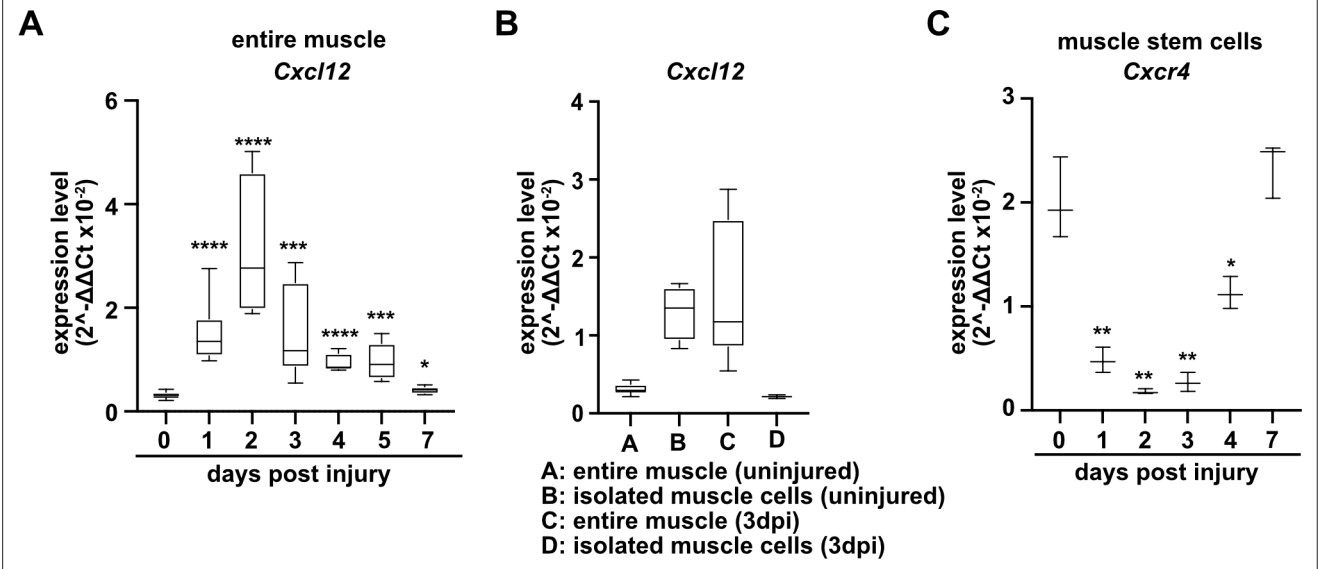

**Figure 4.** Expression of *Cxcl12* and *Cxcr4* during regeneration. (**A**) Expression dynamics of *Cxcl12* in uninjured and regenerating muscle tissue determined by qPCR. (**B**) Expression levels of *Cxcl12* in quiescent and activated muscle stem cells and in muscle tissue during muscle regeneration determined by qPCR. (**C**) Expression levels of *Cxcr4* in quiescent and activated muscle stem cells determined by qPCR. Boxplots represent interquartile range, whiskers show min-to-max range. β-Actin expression was used for normalization in (**A–C**).

The online version of this article includes the following source data and figure supplement(s) for figure 4:

**Source data 1.** Quantification of *Cxcl12* and *Cxcr4* expression represented in the diagrams shown in A-C.

**Figure supplement 1.** Expression levels of *Cxcl12* in quiescent (freshly isolated) and proliferating muscle stem cells at various time points defined by microarray analysis (*Liu et al., 2013*; *Latroche et al., 2017*).

and prolonged inflammation (*Figure 5—figure supplement 2*). In contrast, the single *Cxcr4* mutation in muscle stem cells (*Pax7*[iresCreERT2Gaka/+];*Cxcr4*[flox/flox] treated with tamoxifen, hereafter named Tx[Gaka]*Cxcr4*; *Pax7*[iresCreERT2Gaka/+];*Cxcr4*[+/+] mice treated with tamoxifen served as controls) did neither affect the number of muscle stem cells in regeneration, the diameter of newly formed fibers, nor did it cause prolonged inflammation or fibrosis (*Figure 5*, *Figure 5—figure supplement 1*, and *Figure 5—figure supplement 2*). We conclude that loss of muscle stem cells and deficits in muscle repair are augmented if both *Cxcr4* and *Met* are lacking.

### Muscle stem cells deficient for *Met* and *Cxcr4* are susceptible to apoptosis

We next assessed the mechanisms by which the *Cxcr4* and *Met* mutations affect muscle stem cell maintenance in the injured muscle. We observed a pronounced increase in apoptosis of PAX7+ cells at 4 dpi in the double mutants and a severe decrease in the number of PAX7+ muscle stem cells (*Figure 6*). A less pronounced enhancement of apoptosis of muscle stem cells was observed in Tx[Gaka]*Met* single mutants, whereas the Tx[Gaka]*Cxcr4* single mutation did not significantly impair survival as compared to control animals (*Figure 6*). Thus, the signals provided by CXCR4 and MET protect muscle stem cells from apoptosis in the acutely injured muscle.

CXCR4 and MET signals stimulate muscle stem cell proliferation in vitro (*Allen et al., 1995*; *Cornelison, 2008*). However, in the regenerating muscle in vivo, ablation of *Cxcr4* and *Met* in muscle stem cells did not impair their proliferation. On the contrary, EdU incorporation showed that proliferation of muscle stem cells increased in the Tx[Gaka]*Cxcr4;Met* double and Tx[Gaka]*Met* single mutants (*Figure 6—figure supplement 1*), possibly due to compensatory mechanisms. Moreover, in Tx[Gaka]*Cxcr4;Met* double and Tx[Gaka]*Met* single mutants, the ratio of MyoG+/Pax7+ cells was slightly increased, indicating that differentiation was mildly enhanced (*Figure 6—figure supplement 2*). We conclude that CXCR4 and MET signals cooperate to convey powerful cyto-protective functions.

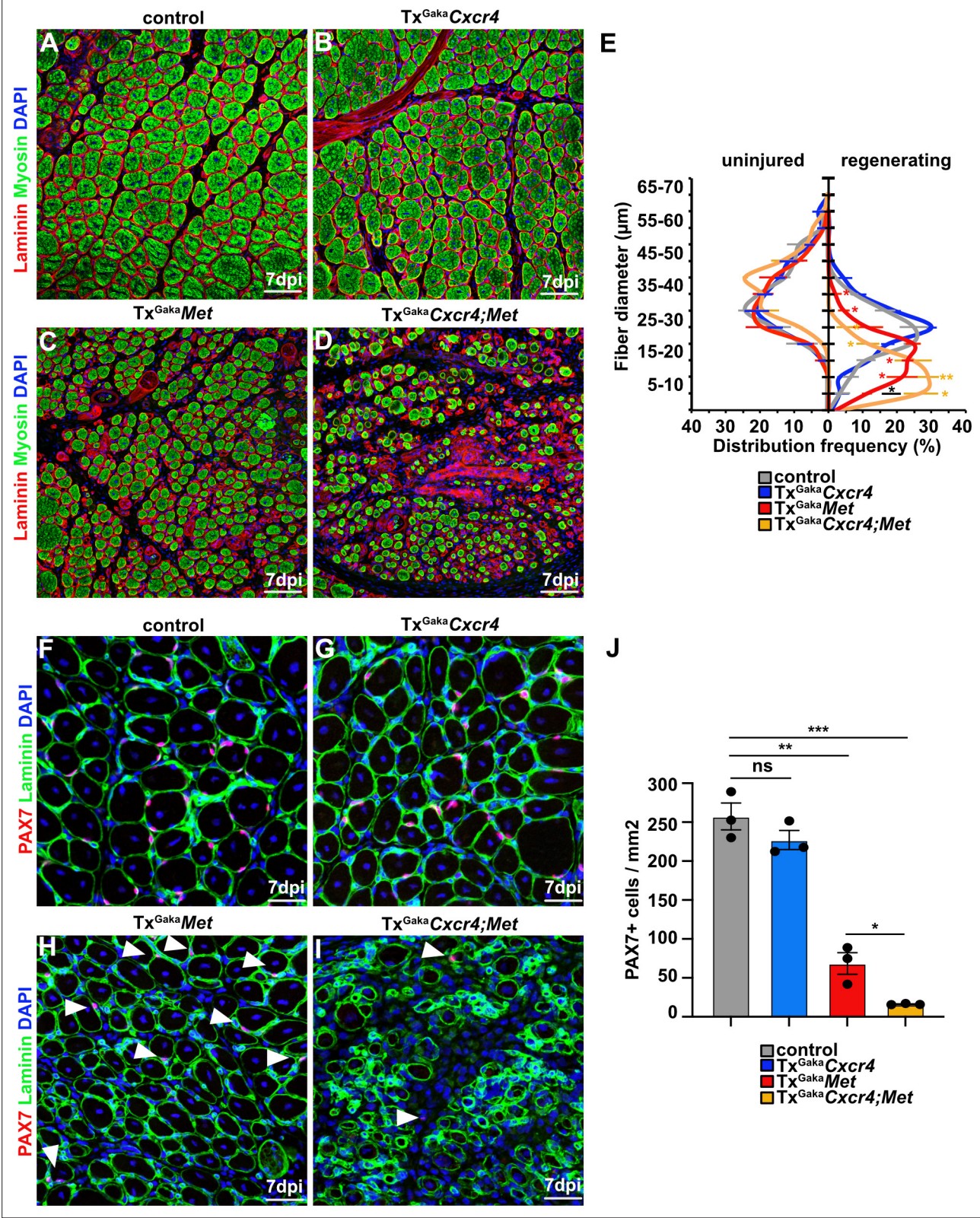

**Figure 5.** *Cxcr4* and *Met* cooperate during muscle regeneration. (**A–D**) Immunohistological analysis of regenerating (7 days post injury [dpi]) muscle of control, Tx^Gaka*Cxcr4*, Tx^Gaka*Met*, and Tx^Gaka*Cxcr4;Met* mice using antibodies against laminin (red) and sarcomeric myosin (green). DAPI was used as a counterstain. Control and mutant animals had been treated with tamoxifen. (**E**) Distribution of Ferret fiber diameters in uninjured and regenerating (7 dpi) muscle of control, Tx^Gaka*Cxcr4*, Tx^Gaka*Met*, and Tx^Gaka*Cxcr4;Met* mice. (**F–I**) Immunohistological analysis of regenerating (7 dpi) muscle of control

*Figure 5 continued on next page*

animals, Tx$^{Gaka}$*Cxcr4*, Tx$^{Gaka}$*Met,* and Tx$^{Gaka}$*Cxcr4;Met* mutants using antibodies against laminin (green) and Pax7 (red). DAPI was used as a counterstain. Arrowheads in (**H**, **I**) point to PAX7+ cells. (**J**) Quantification of PAX7+ cells in regenerating muscle of control, Tx$^{Gaka}$*Cxcr4* and Tx$^{Gaka}$*Met* mice, and Tx$^{Gaka}$*Cxcr4;Met* double mutants. Scale bars, 50 µm (**A–D**), 30 µm (**F–I**). Control: *Pax7$^{iresCreERT2Gaka/+}$*; Tx$^{Gaka}$*Cxcr4: Pax7$^{iresCreERT2Gaka/+}$;Cxcr4$^{flox/flox}$*; Tx$^{Gaka}$*Met: Pax7$^{iresCreERT2Gaka/+}$;Met$^{flox/flox}$*; Tx$^{Gaka}$*Cxcr4;Met: Pax7$^{iresCreERT2Gaka/+}$;Cxcr4$^{flox/flox}$;Met$^{flox/flox}$*. All animals were treated with tamoxifen.

The online version of this article includes the following source data and figure supplement(s) for figure 5:

**Source data 1.** Quantification of fiber diameters, PAX7+ cells and fibrotic area represented in the diagrams shown in E, J (*Figure 5*), E (*Figure 5— figure supplement 1*) and E, F (*Figure 5—figure supplement 2*).

**Figure supplement 1.** Mutations of Cxcr4 and Met in muscle stem cells did not affect muscle stem cell numbers.

**Figure supplement 2.** Increased fibrosis in the regenerating muscle of Met and Cxcr4;Met mutants.

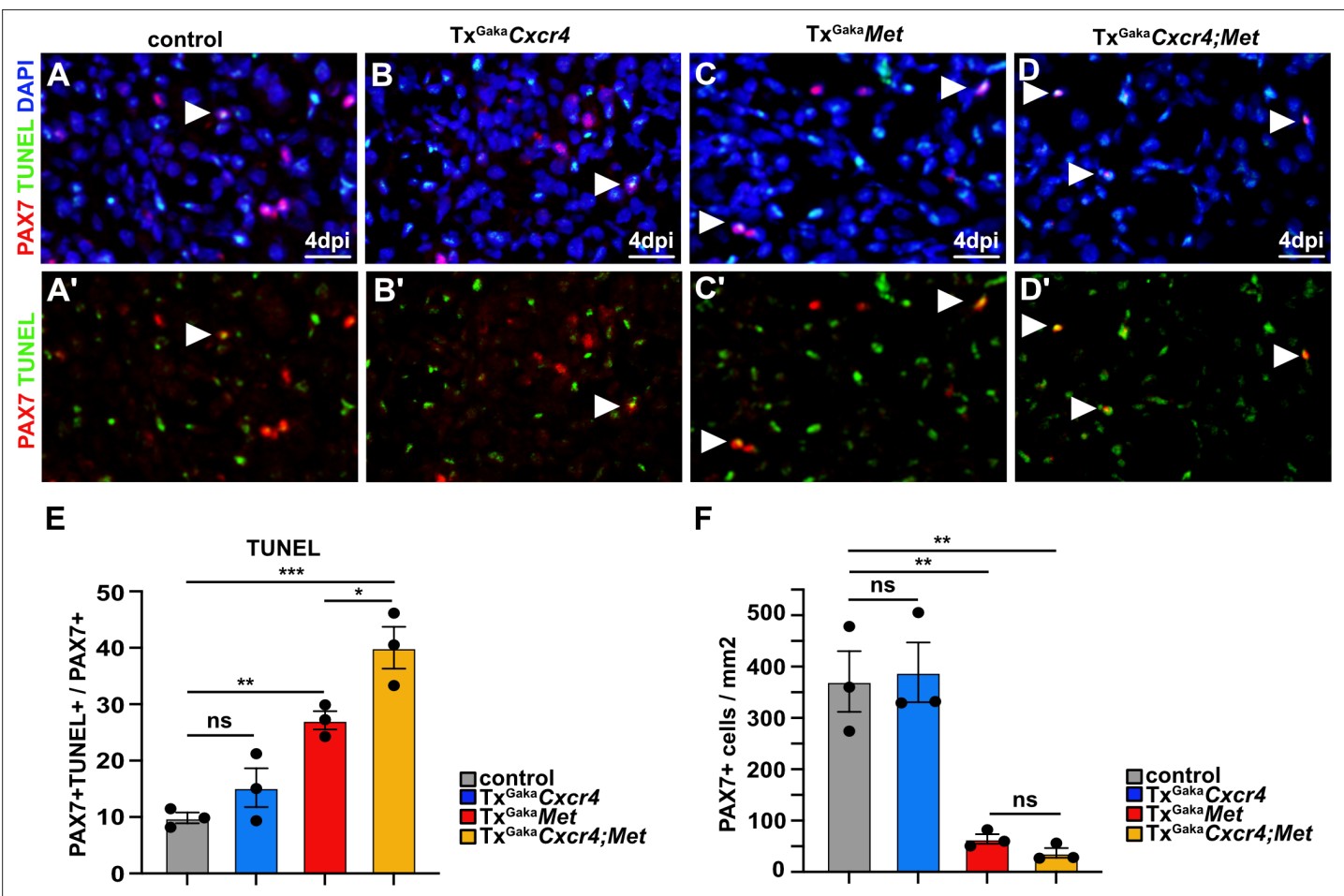

**Figure 6.** *Cxcr4;Met* mutant muscle stem cells undergo apoptosis after acute injury. (**A–D, A'–D'**) Immunohistological analysis of apoptotic cells. PAX7 antibody staining (red) was combined with TUNEL assay (green) to identify apoptotic muscle stem cells in injured muscle of control, Tx$^{Gaka}$*Cxcr4*, Tx$^{Gaka}$*Met*, and Tx$^{Gaka}$*Cxcr4;Met* mice at 4 days post injury (dpi). DAPI was used as a counterstain in (**A–D**). Arrowheads point to TUNEL+ PAX7+ cells. (**E**) Quantification of PAX7+ TUNEL+ cells in regenerating muscle of control, Tx$^{Gaka}$*Cxcr4*, Tx$^{Gaka}$*Met,* and Tx$^{Gaka}$*Cxcr4;Met* mutants. (**F**) Quantification of PAX7+ cells in regenerating muscle of control, Tx$^{Gaka}$*Cxcr4*, Tx$^{Gaka}$*Met*, and Tx$^{Gaka}$*Cxcr4;Met* mice. Scale bars, 20 µm. Control: *Pax7$^{iresCreERT2Gaka/+}$*; Tx$^{Gaka}$*Cxcr4: Pax7$^{iresCreERT2Gaka/+}$;Cxcr4$^{flox/flox}$*; Tx$^{Gaka}$*Met: Pax7$^{iresCreERT2Gaka/+}$;Met$^{flox/flox}$*; Tx$^{Gaka}$*Cxcr4;Met: Pax7$^{iresCreERT2Gaka/+}$;Cxcr4$^{flox/flox}$;Met$^{flox/flox}$*. All animals were treated with tamoxifen.

The online version of this article includes the following source data and figure supplement(s) for figure 6:

**Source data 1.** Quantification of PAX7+TUNEL+ and PAX7+ cells represented in the diagrams shown in E and F (*Figure 6*).

**Figure supplement 1.** Enhanced proliferation of PAX7+ cells in Cxcr4;Met mutants during regeneration.

**Figure supplement 2.** Differentiation is mildly enhanced in Met and Cxcr4;Met mutants during regeneration.

## MET and CXCR4 signaling protects muscle cells from TNFα-induced apoptosis

We next aimed to identify the factor that induces apoptosis of *Met;Cxcr4* mutant muscle stem cells in the injured muscle. The pro-inflammatory cytokine TNFα is induced at the early stages of muscle regeneration and has pro- as well as anti-apoptotic effects on many cell types (*Darnay and Aggarwal, 1999*; *Malka et al., 2000*; *Collins and Grounds, 2001*; *Zador et al., 2001*; *Warren et al., 2002*; *Aggarwal, 2003*). We thus asked whether TNFα production might be responsible for the observed cell death. If freshly isolated muscle stem cells were cultured in media containing 2 % horse serum, TNFα induced apoptosis (*Figure 7A and B*). This TNFα-induced cell death of cultured cells was rescued by the addition of HGF and CXCL12, or by the addition of 10 % fetal calf serum. No cooperative effect of HGF and CXCL12 was observed in this cell culture setting (*Figure 7C and D*).

Finally, using neutralizing antibodies against TNFα,we tested whether the loss of muscle stem cells in the absence of MET and CXCL12 signaling during regeneration in vivo was caused by TNFα. The efficacy TNFα antibodies was verified in a cell culture experiment (*Figure 7—figure supplement 1*). We observed a pronounced rescue of PAX7+ cells in the regenerating muscle of Tx$^{Gaka}$*Met* and Tx$^{Gaka}$*Cxcr4;Met* mutant mice after injection of TNFα neutralizing antibodies (*Figure 7E–P*). Taken together, these data demonstrate that MET and CXCR4 signaling cooperate in vivo to protect muscle stem cells from TNFα-induced apoptosis in the inflammatory environment encountered after injury.

## Discussion

Muscle injury results in an acute inflammatory response causing the recruitment of macrophages and neutrophils. These cells remove cellular debris at the site of injury and provide signals that are beneficial for muscle repair. In addition, they release a multitude of chemokines, as well as reactive oxygen species and enzymes needed to degrade the debris, thereby creating a hostile environment that muscle stems cells have to withstand in order to regenerate the muscle and self-renew (*Tidball, 2005*; *Chazaud et al., 2009*; *Saclier et al., 2013b*; *Londhe and Guttridge, 2015*; *Tidball, 2017*). Our analysis of the in vivo function of MET and CXCR4 demonstrates an important cooperative role in muscle repair that protects stem cells against the adverse environment created by the acute inflammatory response.

Previous studies had shown that HGF can elicit muscle stem cell proliferation in culture and that CXCL12 has mitogenic activity on myogenic C2C12 cells (*Allen et al., 1995*; *Gal-Levi et al., 1998*; *Odemis et al., 2007*). Further, injection of HGF into the intact muscle activates muscle stem cells (*Tatsumi et al., 1998*), and ablation of *Met* in muscle stem cells interferes with entry into G$_{alert}$, a 'alerted' state of quiescence observed in muscle stem cells after injury of the contralateral muscle or of other unrelated organs (*Rodgers et al., 2014*). HGF/MET signaling also affects additional aspects of muscle stem cell biology. In particular, HGF suppresses differentiation of cultured myogenic cell lines and of primary muscle stem cells (*Gal-Levi et al., 1998*; *Siegel et al., 2009*). Thus, HGF had been implicated in multiple aspects of muscle stem cell behaviors, but its role as cyto-protective factor had not been addressed.

Interestingly, cyto-protective functions of HGF/MET were reported in several cell types and injury models, indicating that HGF might be part of a general defensive mechanism in response to tissue damage. In particular, ectopic application of HGF prior to or shortly after an insult protects cells in the liver, kidney, and heart from damage (*Ueda et al., 1999*; *Zhou et al., 2013*; *Matsumoto et al., 2014*; *Pang et al., 2018*). Moreover, after injury to the liver, kidney, heart, or skeletal muscle, increased HGF expression can be observed in the damaged organs, and plasma levels of HGF rise quickly after injury (*Michalopoulos and DeFrances, 1997*; *Nakamura et al., 2000*; *Matsumoto and Nakamura, 2001*). It was proposed that release from extracellular matrix might account for the fast rise in HGF plasma levels (*Shimomura et al., 1995*; *Tatsumi et al., 1998*). In addition, various cytokines, among them interleukin-1 and interleukin-6, activate HGF transcription, which might account for the increased HGF transcripts observed after tissue damage (*Birchmeier et al., 2003*). We demonstrate here that loss of *Met* impairs the resistance of muscle stem cells against acute inflammation. Moreover, in vivo the additional loss of *Cxcr4* exacerbated the deficits observed after loss of *Met*. The cooperative effect that we detected here in vivo is reflected by the fact that both receptors, *Met* and *Cxcr4*, use in part overlapping downstream signaling cascades but also activate distinct signaling molecules. Tyrosine

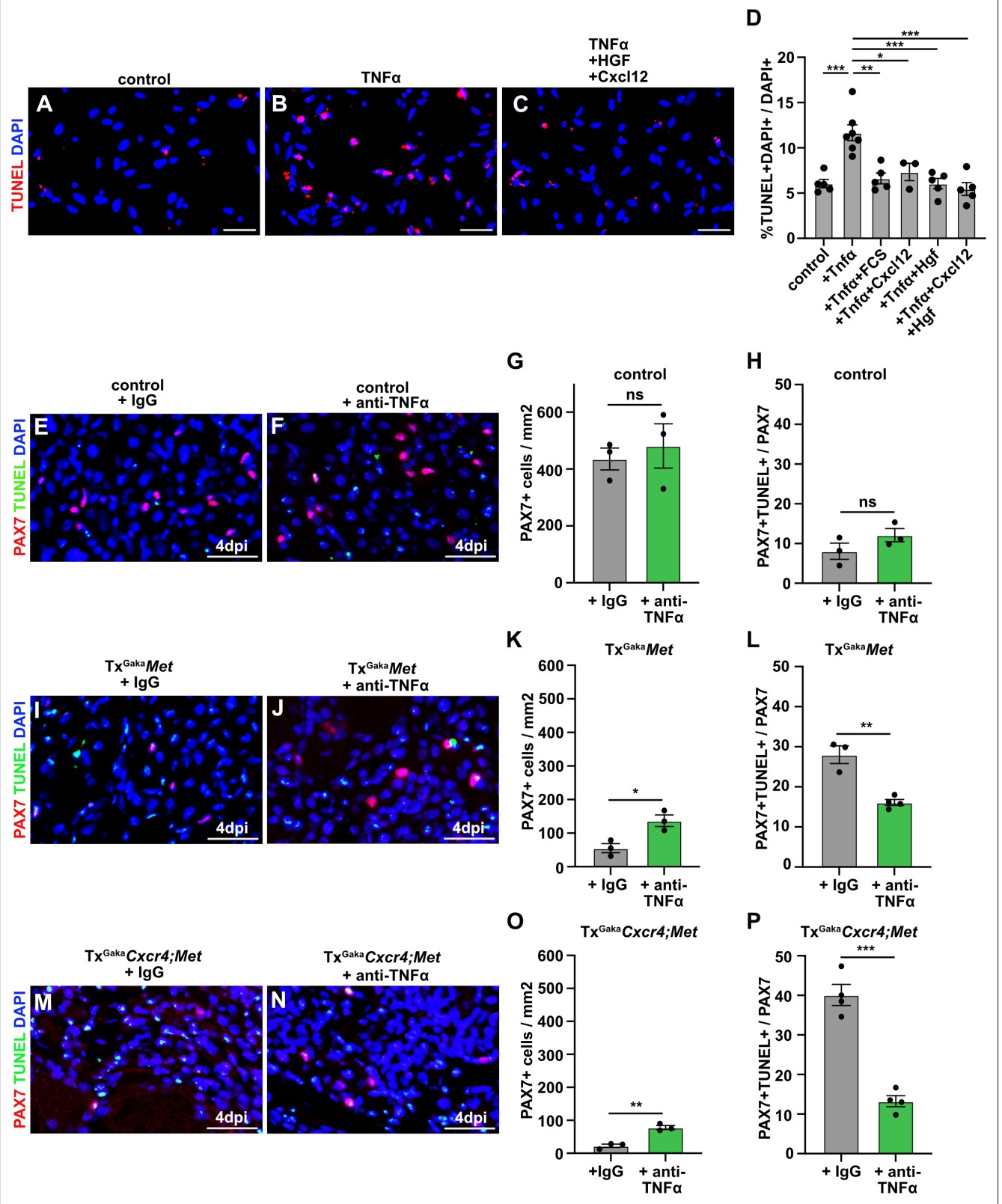

**Figure 7.** CXCL12 and HGF protect muscle cells from TNFα-induced cell death. (**A–C**) Primary muscle stem cells were isolated and cultured for 3 hr in the presence of TNFα plus/minus HGF and Cxcl12. Apoptotic cells were identified by TUNEL staining. (**D**) Quantification of TUNEL+ cells present in such cultures. (**E, F**) Immunohistological analysis of muscle stem cells (PAX7+, red) and apoptotic cells (TUNEL staining, green) in injured muscle (4 days post injury [dpi]) of control mice treated with TNFα neutralizing antibodies or control IgG 2 hr before acute injury. DAPI was used as a

*Figure 7 continued on next page*

*Figure 7 continued*

counterstain. (**G**) Quantification of PAX7+ cells in regenerating muscle (4 dpi) of control mice treated with TNFα neutralizing antibodies or control IgG. (**H**) Quantification of PAX7+ TUNEL+ cells in regenerating muscle (4 dpi) of control mice treated with TNFα neutralizing antibodies or control IgG. (**I, J**) Immunohistological analysis of muscle stem cells (PAX7+, red) and apoptotic cells (TUNEL staining, green) in injured muscle (4 dpi) of Tx$^{Gaka}$*Met* mutants treated with TNFα neutralizing antibodies or control IgG 2 hr before acute injury. DAPI was used as a counterstain. (**K**) Quantification of PAX7+ cells in regenerating (4 dpi) muscle of Tx$^{Gaka}$*Met* mice treated with TNFα neutralizing antibodies or control IgG. (**L**) Quantification of PAX7+ TUNEL+ cells in regenerating muscle from Tx$^{Gaka}$*Met* mice treated with TNFα neutralizing antibodies or control IgG. (**M, N**) Immunohistological analysis of muscle stem cells (PAX7+, red) and apoptotic cells (TUNEL staining, green) in injured muscle (4 dpi) of Tx$^{Gaka}$*Cxcr4;Met* mutants treated with TNFα neutralizing antibodies or control IgG 2 hr before acute injury. DAPI was used as a counterstain. (**O**) Quantification of PAX7+ cells in regenerating muscle (4 dpi) of Tx$^{Gaka}$*Cxcr4;Met* mice treated with TNFα neutralizing antibodies or control IgG. (**P**) Quantification of PAX7+ TUNEL+ cells in regenerating muscle (4 dpi) of Tx$^{Gaka}$*Cxcr4;Met* mice treated with TNFα neutralizing antibodies or control IgG. Scale bars, 20 µm. Control: *Pax7$^{iresCreERT2Gaka/+}$*; Tx$^{Gaka}$*Met*: *Pax7$^{iresCreERT2Gaka/+}$;Met$^{flox/flox}$*; Tx$^{Gaka}$*Cxcr4;Met: Pax7$^{iresCreERT2Gaka/+}$;Cxcr4$^{flox/flox}$;Met$^{flox/flox}$*. All animals were treated with tamoxifen.

The online version of this article includes the following source data and figure supplement(s) for figure 7:

**Source data 1.** Quantification of TUNEL+, PAX7+ and PAX7+TUNEL+ cells represented in the diagrams shown in D, G, H, K, L, O and P (*Figure 7*).

**Figure supplement 1.** Neutralizing capacity of TNFα antibody used in *Figure 7*.

phosphorylation of MET results in the activation of various signaling events that regulate cell motility, proliferation, and survival; among them RAS/MAPK, PI3-kinase/AKT, PLCγ/PKC, RAC/CDC42, and CRK (*Birchmeier et al., 2003*; *Gentile et al., 2008*). CXCR4 uses G-proteins to transmit signals into the cytoplasm, which involves activation of second messenger-regulated serine/threonine kinases or ion channels. However, CXCR4 also activates RAS/MAPK, PI3-kinase/AKT, and CRK signaling, which is particularly well documented in cancer cells (*Teicher and Fricker, 2010*). Among these cascades, PI3-kinase/AKT is well known to act anti-apoptotically, and MAPK/ERK signals can counteract the apoptotic activity of TNFα (*Tran et al., 2001*; *Franke et al., 2003*).

TNFα is one of many pro-inflammatory cytokines that are rapidly induced upon acute muscle injury, and TNFα is highly expressed by pro-inflammatory macrophages. The primary role of TNFα is to regulate immune cells, but it also affects the proliferation and differentiation of cultured muscle cells (*Wallach et al., 1999*; *Li, 2003*; *Luo et al., 2005*; *Palacios et al., 2010*). Mice lacking TNFα receptors p55 and p75 show that TNFα does not play an essential role in muscle regeneration, indicating that this cytokine seems to act redundantly with other factors (*Collins and Grounds, 2001*). However, systemic injection of TNFα neutralizing antibodies protected dystrophic skeletal muscle of *mdx* mice from necrosis and increased the number of PAX7+ cells (*Palacios et al., 2010*). This indicates that TNFα exacerbates muscle fiber damage and, in addition, impairs muscle stem cell maintenance in dystrophic muscle. Our analysis indicates that TNFα signals are also damaging for muscle stem cells during acute inflammation after injury, but that endogenous HGF and CXCL12 may counteract this. Effects of TNFα are modulated by other signals, and particularly MAPK/ERK activity can override the apoptotic TNFα signal (*Tran et al., 2001*; *Aggarwal, 2003*; *Wada and Penninger, 2004*; *Lu and Xu, 2006*; *Lau et al., 2011*). Acute skeletal muscle injury resulting in inflammation is a common clinical condition caused by trauma, severe contraction, chemicals, myotoxins, and ischemia. Similarly, acute inflammation is observed in muscle diseases like dystrophy (*Kharraz et al., 2014*; *Tidball et al., 2018*). Our genetic experiments indicate that HGF/MET and CXCL12/CXCR4 signaling protects muscle stem cells against the noxious environment generated by the inflammatory response. Exogenous HGF was previously tested in muscle injury and increased the numbers of activated muscle stem cells, but did not enhance fiber growth (*Miller et al., 2000*). Thus, in healthy muscle, endogenous factors, among them HGF, suffice to ensure appropriate regeneration. Nevertheless, in muscle disease where repair mechanisms fail, enhanced cyto-protection of muscle stem cells appears to be beneficial (*Palacios et al., 2010*). Whether HGF/MET and CXCL12/CXCR4 signaling protects against TNFα-induced damage in such disease settings will need further investigation.

## Materials and methods

### Key resources table

| Reagent type (species) or resource | Designation | Source or reference | Identifiers | Additional information |
|---|---|---|---|---|
| Antibody | Guinea pig polyclonal anti-PAX7 | Our lab | PMID:22940113 | 1:2500 |

*Continued on next page*

| Reagent type (species) or resource | Designation | Source or reference | Identifiers | Additional information |
|---|---|---|---|---|
| Antibody | Rabbit polyclonal anti-Laminin | Sigma-Aldrich | L9393 RRID:AB_477163 | 1:500 |
| Antibody | Goat polyclonal anti-CollagenIV | Millipore | AB769 RRID:AB_92262 | 1:500 |
| Antibody | Mouse monoclonal anti-sarcomeric myosin | DSHB | MF20 RRID:AB_2147781 | 1:10 |
| Antibody | Rabbit polyclonal anti-Myogenin | Abcam | ab124800 RRID:AB_10971849 | 1:1000 |
| Antibody | Mouse monoclonal anti-F4/80 | Abcam | ab6640 RRID:AB_1140040 | 1:100 |
| Antibody | Rabbit polyclonal anti-fibronectin | Sigma-Aldrich | F7387 RRID:AB_476988 | 1:500 |
| Antibody | Cy2, Cy3, Cy5 conjugated antibodies | Dianova | | 1:500 |
| Commercial assay or kit | In Situ Cell Death Detection Kit | Roche | 12156792910 | |
| Commercial assay or kit | EdU | baseclick GmbH | BCK-EdU647 | |
| Commercial Assay or kit | qPCR SYBR Green Mix | Thermo Fisher | AB1158B | |
| Sequence-based reagent | ATCCACGATGTTCATGAGAG | Eurofins | N/A | qPCR HGF (forward primer) |
| Sequence-based reagent | GCTGACTGCATTTCTCATTC | Eurofins | N/A | qPCR HGF (reverse primer) |
| Sequence-based reagent | CACAGAAAGCATGATCCGCGACGT | Eurofins | N/A | qPCR TNF (forward primer) |
| Sequence-based reagent | CGGCAGAGAGGAGGTTGACTTTCT | Eurofins | N/A | qPCR TNF (reverse primer) |
| Sequence-based reagent | CAGAGCCAACGTCAAGCA | Eurofins | N/A | qPCR Cxcl12 (forward primer) |
| Sequence-based reagent | AGGTACTCTTGGATCCAC | Eurofins | N/A | qPCR Cxcl12 (reverse primer) |
| Sequence-based reagent | CATTTTGGCTGTGTCTATCATG | Eurofins | N/A | qPCR Met (forward primer) |
| Sequence-based reagent | ACTCCTCAGGCAGATTCCC | Eurofins | N/A | qPCR Met (reverse primer) |
| Sequence-based reagent | TCAGTGGCTGACCTCCTCTT | Eurofins | N/A | qPCR CXCR4 (forward primer) |
| Sequence-based reagent | CTTGGCCTTTGACTGTTGGT | Eurofins | N/A | qPCR CXCR4 (reverse primer) |
| Sequence-based reagent | CATTTTGGCTGTGTCTATCATG | Eurofins | N/A | qPCR Met Exon 17 (forward primer) |
| Sequence-based reagent | ACTCCTCAGGCAGATTCCC | Eurofins | N/A | qPCR Met Exon 18 (reverse primer) |
| Sequence-based reagent | CTTGCCAGAGACATGTACGAT | Eurofins | N/A | qPCR Met Exon 20 (forward primer) |
| Sequence-based reagent | AGGAGCACACCAAAGGACCA | Eurofins | N/A | qPCR Met Exon 21 (reverse primer) |
| Sequence-based reagent | CCAGTTGGTAACAATGCCATGT | Eurofins | N/A | qPCR β-actin (forward primer) |
| Sequence-based reagent | GGCTGTATTCCCCTCCATCG | Eurofins | N/A | qPCR β-actin (reverse primer) |
| Sequence-based reagent | ACTAGGCTCCACTCTGTCCTTC | Eurofins | PMID:19554048 | Genotyping PCR-Primer 1 Pax7CreERT2Fan |
| Sequence-based reagent | GCAGATGTAGGGACATTCCAGTG | Eurofins | PMID:19554048 | Genotyping PCR-Primer 2 Pax7CreERT2Fan |
| Sequence-based reagent | GCTGCTGTTGATTACCTGGC | Eurofins | PMID:21828091 | Genotyping PCR-Primer 1 Pax7CreERT2GaKa |
| Sequence-based reagent | CTGCACTGAGACAGGACCG | Eurofins | PMID:21828091 | Genotyping PCR-Primer 2 Pax7CreERT2GaKa |

| Reagent type (species) or resource | Designation | Source or reference | Identifiers | Additional information |
|---|---|---|---|---|
| Sequence-based reagent | GCTGCTGTTGATTACCTGGC | Eurofins | PMID:21828091 | Genotyping PCR-Primer 1 Pax7CreERT2GaKa |
| Sequence-based reagent | GCTCTGGATACACCTGAGTCT | Eurofins | PMID:15520281 | Genotyping PCR-Primer 1 Pax7-IRESCre |
| Sequence-based reagent | GGATAGTGAAACAGGGGCAA | Eurofins | PMID:15520281 | Genotyping PCR-Primer 2 Pax7-IRESCre |
| Sequence-based reagent | TCGGCCTTCTTCTAGGTTCTGCTC | Eurofins | PMID:15520281 | Genotyping PCR-Primer 3 Pax7-IRESCre |
| Sequence-based reagent | CCACCCAGGACAGTGTGACTCTAA | Eurofins | PMID:15520246 | Genotyping PCR-Primer 1 Cxcr4 flox |
| Sequence-based reagent | GATGGGATTCTGTATGAGGATTAGC | Eurofins | PMID:15520246 | Genotyping PCR-Primer 2 Cxcr4 flox |
| Sequence-based reagent | CCAAGTGTCTGACGGCTGTG | Eurofins | N/A | Genotyping PCR-Primer 1 Met flox |
| Sequence-based reagent | AGCCTAGTGGAATTCTCTGTAAG | Eurofins | N/A | Genotyping PCR-Primer 2 Met flox |

## RNA isolation and qPCR

RNA from the entire muscle and from FACS-isolated muscle stem cells was extracted using TRIzol reagent (15596026, Thermo Fisher Scientific) following the manufacturer's instructions. qPCR was performed using SYBR green master mix (4309155, Thermo Fisher Scientific) as described previously (*Bröhl et al., 2012*). PCR primers are listed in Key resources table. β-Actin was used for normalization.

## Immunohistochemistry

Immunohistochemistry was performed on 12 µm cryo-sections of muscle biopsy samples fixed in Zamboni's fixative for 20 min as described previously (*Bröhl et al., 2012*). For staining of Pax7, sections were incubated in Antigen Unmasking Solution buffer (H-3300, Vector Laboratories) for 20 min at 80 °C. Primary and secondary antibodies used are listed in Key resources table. Primary antibodies were incubated overnight, and secondary antibodies for 1 hr at 4 °C in blocking solution. DAPI (D9542, Sigma-Aldrich) was used as a counterstain to label nuclei. To detect apoptotic cells, Pax7 immunohistochemistry was combined with TUNEL TMR Red detection kit according to the manufacturer's instruction (12156792910, Roche). To monitor proliferating cells, EdU (50 µg/g body weight) was given i.p. 2 hr before the isolation of the muscle. EdU was detected using Click chemistry (EdU-Click 647, BCK-EdU647, baseclick GmbH) and Biotin picolyl azide (900912, Sigma-Aldrich) as substrate. Detection was performed with fluorophore-coupled streptavidin. Images were acquired using a LSM700 confocal microscope and processed using Adobe Photoshop (Adobe Systems).

## Isolation of muscle stem cells and muscle injury

Muscle stem cells were isolated from skeletal muscle using fluorescent-activated cell sorting (FACS) as described (*Bröhl et al., 2012*). Shortly, muscle tissue was minced, enzymatically digested with 1.5 U/ml NB4G Collagenase (S1745401, Serva), and 2.4 U/ml Dispase (04942078001, Sigma-Aldrich). Mononucleated cells were isolated and labeled with antibodies against VCAM, Sca1, CD45, CD31 (AF643, rndsystems; BD Bioscience). VCAM+ Sca1 CD31-CD45-cells were isolated using a BD Aria III sorter (BD Bioscience) and dead cells were excluded by propidium iodide staining (P4864, Sigma-Aldrich). Muscle stem cells from regenerating muscles were isolated from animals carrying Pax7$^{nGFP}$ allele using the digestion procedure described above. Mono-nucleated cells GFP+ cells were isolated by FACS. Cells were collected in TRIzol RNA extraction reagent (15596026, Thermo Fisher Scientific) for RNA isolation or in DMEM/10 % FCS for cultivation.

Muscle injury was induced by injecting 30 µl of cardiotoxin (10 µM; C9759, Sigma-Aldrich) into the *tibialis anterior* muscle of 8- to 12-week-old mice. Muscle injected with phosphate buffered saline (10010056, Thermo Fisher Scientific) was used as a control. Recombination using *CreERT2* alleles was induced as described (*Murphy et al., 2011*), and the injury was induced 10 days after the last tamoxifen administration. Antigen affinity-purified polyclonal goat human/mouse TNFα antibody (AF-410-NA, rndsystems, LOT NQ2519111, NQ2520111, NQ2418041) was dissolved in PBS and

100 µg were injected in a single injection i.p. 2 hr before the cardiotoxin injection. Mice injected with 100 µg goat IgG (AB-108-C, rndsystems) served as control. The animals were analyzed 4 days after injury.

## Mouse strains

The Cxcr4flox, Metflox, Pax7nGFP, Pax7$^{iresCre}$, Pax7$^{CreERT2Fan}$ , and Pax7$^{iresCreERT2Gaka}$ mouse strains have been described previously (**Borowiak et al., 2004**; **Keller et al., 2004**; **Nie et al., 2004**; **Lepper et al., 2009**; **Sambasivan et al., 2009**; **Murphy et al., 2011**). Heterozygous animals carrying the Pax7$^{iresCre}$ allele served as controls for Pax7$^{iresCre/+}$;Met$^{flox/flox}$ (coMet) mutants. Heterozygous animals carrying the Pax7$^{iresCreERT2Gaka}$ or Pax7$^{CreERT2Fan}$ treated with tamoxifen served as controls for Pax7$^{iresCreERT2Gaka/+}$;Met$^{flox/flox}$ (Tx$^{Gaka}$Met) and Pax7$^{CreERT2Fan/-}$;Met$^{flox/flox}$ (Tx$^{Fan}$Met) mutants, respectively, in all experiments but those shown in **Figure 3—figure supplement 1A–D**, where Cre-negative littermates served as controls. Mice were maintained on a mixed 129/Sv and C57BL/6 genetic background. All experiments were conducted according to regulations established by the Max-Delbrück-Center for Molecular Medicine (MDC) and the Landesamt für Gesundheit und Soziales, Berlin (0320/10; 0130/13).

## Cultivation, induction of apoptosis, and rescue of muscle stem cells in culture

Neutralization capacity of the TNFα antibody (AF-410-NA, rndsystems) was tested in vitro. C2C12 cells (ATCC, CRL-1772; not listed by ICLAC) in DMEM (1196508, Thermo Fisher Scientific) containing 2 % horse serum (16050122, Thermo Fisher Scientific), 1 % Penicillin/Streptomycin (15140122, Thermo Fisher Scientific), and 1 % GlutaMax (35050061, Thermo Fisher Scientific) were exposed to 60 ng/ml recombinant TNFα (210-TA, rndsystems) and different concentrations of TNFα neutralizing antibody (60 ng/ml, 180 ng/ml, 600 ng/ml, and 1800 ng/ml) overnight, fixed in 4 % paraformaldehyde (PFA), and apoptotic cells were detected using TUNEL TMR Red detection kit according to the manufacturer's instruction (12156792910, Roche). The ratio of TUNEL+ DAPI+/DAPI+ cells was quantified in randomly chosen areas of triplicate experiments using a LSM700 Zeiss confocal microscope and ImageJ 'cell counter' plug-in for quantification. The cell identity of the C2C12 cell line used in this study was tested by in vitro differentiation into multinuclear myotubes. Differentiation was achieved by replacing growth media (GM) ( 10 % fetal calf serum (FCS) F7524, Sigma-Aldrich, DMEM 1196508, Thermo Fisher Scientific, 1 % Penicillin/Streptomycin 15140122, Thermo Fisher Scientific, 1 % GlutaMax 35050061, Thermo Fisher Scientific) to differentiation media (DM) (2 % horse serum 16050122, Thermo Fisher Scientific, DMEM 1196508, Thermo Fisher Scientific, 1 % Penicillin/Streptomycin 15140122, Thermo Fisher Scientific, 1 % GlutaMax 35050061, Thermo Fisher Scientific). Formation of myotubes was observed 4 days after replacing the GM to DM. Differentiation was confirmed by immunohistochemistry using an antibody against Myogenin. The C2C12 cell line was tested negative for mycoplasma contamination.

FACS-isolated muscle stem cells were cultivated on 10 % Matrigel (354230, Corning Life Sciences) in DMEM/F-12 (11320074, Thermo Fisher Scientific) containing 10 % fetal calf serum (F7524, Sigma-Aldrich), 5 % horse serum (16050122, Thermo Fisher Scientific), 0.1 % bovine FGF (F5329, Sigma-Aldrich), 1 % Penicillin/Streptomycin (15140122, Thermo Fisher Scientific), and 1 % GlutaMax (35050061, Thermo Fisher Scientific) for 24 hr, and subsequently incubated in DMEM/F-12 containing 2 % horse serum. Recombinant human TNFα (210-TA, rndsystems) was added to a final concentration of 120 ng/ml, and cell survival was assayed 3 hr later. Recombinant Cxcl12 (250-20A, Peprotech) and HGF protein (kindly provided by W. Birchmeier) were used at final concentrations of 20 ng/ml and 25 ng/ml, respectively. After 3 hr incubation, cells were fixed in 4 % PFA and washed twice with phosphate-buffered saline (PBS) (10010056, Thermo Fisher Scientific). To detect apoptotic cells, Pax7 immunohistochemistry was combined with TUNEL TMR Red detection according to the manufacturer's instruction (12156792910, Roche); DAPI (D9542, Sigma-Aldrich) was used as a counterstain. The ratio of TUNEL+ DAPI+/DAPI+ cells was quantified in randomly chosen areas of three different experiments using a LSM700 Zeiss confocal microscope and ImageJ 'cell counter' plug-in.

## Computational analysis and statistics

Gene expression levels of freshly isolated and cultured muscle stem cells were previously determined using gene expression microarrays (**Liu et al., 2013**; **Latroche et al., 2017**). The *.CEL files of scanned

Affymetrix mRNA expression microarrays were downloaded from the GEO repository (accession codes GSE47177 and GSE103684, n = 3 replicates/condition). Normalization and background corrections were performed using the AffySTExpressionCreator v0.14 on the GenePattern Server (*Reich et al., 2006*) running the Robust Multi-array Average (RMA) algorithm (*Irizarry et al., 2003*). The relative signal intensities of gene expression of muscle stem cell activation were plotted against the time axis.

Three or more animals were used per genotype and experiment. Microsoft Excel and GraphPad Prism 9 were used for statistical analysis. Data were analyzed using an unpaired, two-tailed t-test. p-values < 0.05 were considered significant. Results are shown as arithmetical mean ± standard error of the mean (SEM) and the dots represent the mean of individual animals. ns: not significant, $p > 0.05$, $*p < 0.05$, $**p < 0.01$, $***p < 0.001$.

## Acknowledgements

We thank Walter Birchmeier, Thomas Müller, and Minchul Kim for helpful discussions. We are grateful to Vivian Schulz, Pia Blessin, and Sven Buchert for technical assistance, and to Petra Stallerow and Claudia Päseler for help with the animal husbandry. We also acknowledge Elijah Lowenstein and Thomas Müller for critically reading the manuscript. This work was supported by the Deutsche Forschungsgemeinschaft (DFG, Klinische Forschergruppe KFO 192 and AFM/Telethon to CB).

## Additional information

### Competing interests

Jie-Shin Chen: Jie-Shin Chen is now affiliated with AstraZeneca; all work for this manuscript was conducted while affiliated with Max Delbrueck Center for Molecular Medicine (MDC) in the Helmholtz Society. The other authors declare that no competing interests exist.

### Funding

| Funder | Grant reference number | Author |
| --- | --- | --- |
| Deutsche Forschungsgemeinschaft | | Carmen Birchmeier |
| AFM | | Carmen Birchmeier |
| Klinische Forschergruppe KFO 192 | | Carmen Birchmeier |

The funders had no role in study design, data collection and interpretation, or the decision to submit the work for publication.

### Author contributions

Ines Lahmann, Conceptualization, Investigation, Writing – original draft; Joscha Griger, Jie-Shin Chen, Yao Zhang, Investigation; Markus Schuelke, Computational analysis; Carmen Birchmeier, Conceptualization, Supervision, Writing – original draft

### Author ORCIDs

Carmen Birchmeier (iD) http://orcid.org/0000-0002-2041-8872

### Ethics

All experiments were conducted according to regulations established by the Max-Delbrück- Center for Molecular Medicine (MDC) and the Landesamt für Gesundheit und Soziales (0320/10; 0130/13).

### Decision letter and Author response

Decision letter https://doi.org/10.7554/eLife.57356.sa1
Author response https://doi.org/10.7554/eLife.57356.sa2

## Additional files

### Supplementary files
• Supplementary file 1. *Tnf*, *Hgf,* and *Cxcl12* expression levels during muscle regeneration. Expression levels of *Tnf*, *Hgf,* and *Cxcl12* mRNA after acute injury were determined in the entire muscle by qPCR. Uninjured and 1–7 days post injury (dpi) were assessed, and expression was normalized to the expression in the uninjured muscle. The values are displayed as means ± SEM. p-Values are shown in brackets. β-Actin was used for normalization.

• Transparent reporting form

### Data availability
All data generated or analysed during this study are included in the manuscript and supporting files.

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
