## [Decision Letter]

**Acceptance summary:**

MET/HGF and CXCR4/CXCL12 signals have long been known to play roles in muscle stem cells, but in this manuscript the authors demonstrate a new role for MET and CXCR4. Using conditional mutagenesis and rescue experiments, they show that MET and CXCR4 act cooperatively during regeneration to protect stem cells from apoptosis, via inhibition of the pro-inflammatory cytokine TNFα.

**Decision letter after peer review:**

Thank you for submitting your article "Met and Cxcr4 signals cooperate to protect muscle stem cells against inflammation-induced damage during regeneration" for consideration by *eLife*. Your article has been reviewed by 3 peer reviewers, including Gabrielle Kardon as the Reviewing Editor and Reviewer #1, and the evaluation has been overseen by Didier Stainier as the Senior Editor. The following individual involved in review of your submission has agreed to reveal their identity: So-ichiro Fukada (Reviewer #2).

The reviewers have discussed the reviews with one another and the Reviewing Editor has drafted this decision to help you prepare a revised submission.

Summary:

Lahmann and colleagues examine in "Met and Cxcr4 signals cooperate to protect muscle stem cells against inflammation-induced damage during regeneration" the cell-autonomous role of Met and Cxcr4 signaling in satellite cells during regeneration. The role of Met signaling, in particular, in muscle regeneration has been examined by multiple papers over many years, and roles in activation, proliferation, migration, fusion, and quiescence have all been implicated. The novel finding of this paper is that the authors find that Met and Cxcr4 in satellite cells protects these cells from apoptosis early in regeneration (3 days post injury) in response to TNFα expression. They also suggest that Met and Cxcr4 act cooperatively and HGF/Met and CXCL12/Cxc4 signaling act in an autocrine manner to protect satellite cells. The reviewers were generally enthusiastic about the proposed new role of Met and Cxcr4 in protecting satellite cells from apoptosis during regeneration. The reviewers found that manuscript is well written and the experimental approach is overall of high-standards, involving a large number of genetic models. However, there were several deficiencies noted (particularly with respect to controls) that need to be addressed. The data supporting that HGF/Met and CXCL12/Cxcr4 functions in an autocrine manner was found to be less well supported and in need of more experimental data to be included in the manuscript.

Essential revisions:

1. A major concern is the issue of the experimental mice used in these experiments. For most experiments Pax7iresCre mice were used to delete Met and/or Cxcr4. As Pax7iresCre deletes in muscle progenitors during development, both the satellite cells and myofibers have Met and Cxcr4 deleted at the start of the regeneration experiments and thus the experiments are not strictly testing the role of Met and/or Cxcr4 only during muscle regeneration. The authors have addressed this in a small number of experiments in Figure 1 and Figure S1. They show that the cross-sectional area of myofibers and the number of satellite cells does not differ between control and Pax7iresCre uninjured muscle, suggesting that loss of Met in Pax7+ cells does not have a major developmental defect. In addition they show (Figure 1K and S1B-E) that in response to injury the number of Pax7+ cells is reduced 7dpi with deletion in Pax7iresCre as well as Pax7iresCreERT2GAKA and Pax7CreERT2FAN mice. Thus they argue that the results with the Pax7iresCre are similar to the results with the tamoxifen-inducible Pax7CreERT2 mice and reflect the role of Met during regeneration and not during development. However, the results in Figure 1K and Figure S1E are not able to be compared because they use different metrics (Pax7+ cells/100 myofibers vs Pax7+ cells/area) and the identity of controls are unclear. Thus whether the results with the Pax7iresCre really reflect the requirement of Met and/or Cxcr4 strictly during regeneration is uncertain. The authors need to repeat key experiments using the Pax7CreERT2FAN or Pax7CreERT2GAKA mice.

2. There is concern about whether the appropriate control mice have been used in the genetic experiments. Throughout all figure panels, the full genotype of all control and experimental mice should be displayed. Particularly in experiments using Pax7CreERT2FAN, the control mice should be Pax7CreERT2Fan/+; Met+/+ and with tamoxifen. The issue of the control mice used was particularly of concern in Figure S1E.

3. The authors conclude that HGF/Met and CXCL12/Cxcr4 signaling is autocrine in satellite cells. However, they do not provide enough data to support such a conclusion. They show in Figures 1D and 2A by smFISH that HGF and CXCl12 is co-expressed with Pax7 in satellite cells, although there was some concern about the quality of this data. The expression of HGF and CXCL12 in satellite cells could be strengthen by examining their expression via qPCR in isolated satellite cells rather than in whole muscle homogenates. Even with such data, the authors can only suggest, but not conclude, that HGF and CXCL12 function in an autocrine manner. HGF and CXCL12 from other cell types may be critical. A definitive test would require that HGF or CXCL12 are conditionally deleted in satellite cells via Pax7CreERT2.

4. The authors propose that Met and Cxcr4 act cooperatively to prevent TNFa-mediated apoptosis. While the authors show the number of apoptotic Pax7+ cells is increased in Pax7iresCre/+; Metfl/fl; Cxcr4fl/fl (Figure 3C), they do not quantify the number of apoptotic Pax7+ cells in Pax7iresCre/+; Metfl/fl or Pax7iresCre/+; Cxcr4fl/fl. Also the control genotype for these experiments is not detailed. The data for all four genotypes needs to be included in order for a role of Met and Cxcr4 cooperativity to be assessed.

5. Figure 4A-C shows an increase in propidium iodide+ satellite cells cultured in the presence of TNFa, which is rescued when either HGF, CXCL12, or HGF and CXCL12 are added. Propidium iodide is an assay for nonviable cells. The authors should conduct this experiment with TUNEL assay (as in Figure 3A-C). In addition, these data suggest that either HGF or CXCL12 are sufficient to rescue cell death and there is no additive benefit to using both HGF and CXCL12. This does not support the contention that HGF and CXCL12 are both required to protect satellite cells from TNFα -induced apoptosis.

6. The in vivo role of Met and Cxcr4 in protection against TNFa -induced apoptosis needs to be strengthened. The authors need to show in vivo whether loss of Met alone leads to an increase in satellite cell apoptosis at 3 dpi. Also, the rescue experiments using neutralizing TNFa antibody (Figure 4D-F) only assay the number of Pax7+ cells/area and not changes in numbers of apoptotic Pax7+ cells; this should be included.*Reviewer #1:*

Lahmann and colleagues examine in "Met and Cxcr4 signals cooperate to protect muscle stem cells against inflammation-induced damage during regeneration" the cell-autonomous role of Met and Cxcr4 signaling in satellite cells during regeneration. The role of Met signaling, in particular, in muscle regeneration has been examined by multiple papers over many years, and roles in activation, proliferation, migration, fusion, and quiescence have all been implicated. The novel finding of this paper is the implication that Met and Cxcr4 protects satellite cells from apoptosis early in the regenerative response (3 days post injury) due to TNFa expression. They also suggest that Met and Cxcr4 act cooperatively and in an autocrine manner to protect satellite cells. The finding of a role for Met and Cxcr4 for cooperatively blocking apoptosis during regeneration is interesting, but in need of further data to support. There is little data in this to support that HGF/Met and CXCL12/Cxc4 functions in an autocrine manner. See specific comments.

1. The authors show by qPCR that satellite cells express Met and by smFish that some satellite cells express HGF and conclude "HGF I (probably a typo – "is") produced by muscle stem cells and functions in an autocrine manner during repair" p.5. Based on their expression data and conditional deletion in satellite cells, they can not conclude this. They show that Met is required in satellite cells, but the source of HGF may be from satellite cells or many other cell types present in regenerating muscle that they have not tested.

2. Figure 1E-L: In these experiments they have deleted Met using the Pax7iresCre, and so Met has been deleted in muscle progenitors throughout development. They need to explicitly state this in the last paragraph of p. 5. I find it surprising that there is no developmental phenotype (neither number of Pax7+ muscle progenitors or myofiber cross-sectional area is affected). Please make sure to highlight this in text. Also, please put the actual genotype of control and experimental mice on the Figure panels – do not use the abbreviation "control" and "coMet"; we need to see the actual genotypes of these mice. The authors state that "similar deficits were observed when Met was mutated in adult muscle stem cells using the tamoxifen-inducible Pax7iresCreERT2GAKA allele". Only the reduction in Pax7+ satellite cells at 7 dpi is shown and not the changes in myofiber cross-sectional area; this should be shown.

3. The authors write on p. 6 that "loss of Met in muscle stem cells results in a mild regeneration deficit that is accompanied by a reduction in the number of muscle stem cells". However, the authors never analyze any muscle regeneration phenotypes after 7 dpi. Certainly the work of Webster and Fan 2013 shows a severe regeneration phenotype in the myofibers at later time points (20 dpi). The authors need to look at later time points post injury or explicitly acknowledge the work of Webster and Fan, which clearly shows a regeneration defect.

4. Figure 2: The case of cooperativity between Met and Cxcr4 would be made easier to see if they included the data on Pax7iresCre;Cxcr4fl/fl mice in the main figure and not in Figure S2. It is unclear if there is increased fibrosis if both Met and Cxcr4 are deleted, versus individual loss of Met and Cxcr4. If they want to make this point they need to include all 4 genotypes (Control that is specified; Pax7iCre; Cxcr4fl/fl; Pax7iCre;Metfl/fl; and Pax7iCre;Cxcr4fl/fl;Metfl/fl) and quantify the degree of fibrosis.

5. Figure 3: Panels A-C. The authors show an increase in the number of Tunel+Pax7+ cells at 3dpi in Pax7iCre; Cxcr4fl/fl;Metfl/fl mice. The authors need to show the quantification of Tunel+Pax7+ cells at 3 dpi for Pax7iCre; Cxcr4fl/fl and Pax7iCre;Metfl/fl mice. Also please write the genotype of the "control" mice. Panels D-H. The authors need to show the number of Pax7+ and BrdU^+^Pax7+ cells for all four genotypes: 1. control, which needs to be specified; 2. Pax7iCre; Cxcr4fl/fl; 3. Pax7iCre;Metfl/fl; and 4. Pax7iCre;Cxcr4fl/fl;Metfl/fl. Without all four genotypes, it is not possible to infer whether the effects of Met and Cxcr4 really are cooperative.

6. Figure 4: 1. The authors show in Panel C that either Cxcl12 or HGF alone is sufficient to rescue TNFa induced satellite cell death in culture and the effects of Cxcl12 and HGF do not lead to a further rescue. Thus it is most parsimonious to argue that Cxcl12 and HGF do not have a synergistic (cooperative) or additive effect on rescue of cell death – either factor will work. 2. Panel D. The authors show that blocking TNFa partially rescues the number of Pax7+ cells when Cxcr4 and Met are deleted in Pax7+ cells. It is important for the authors to show the effects of TNFa blockade on Pax7iCre; Cxcr4fl/fl and Pax7iCre;Metfl/fl, if there is increased apoptosis in these genotypes (see comments above).

7. The authors claim that the "major role" of Met and Cxcr4 is to "work together in order to protect stem cells against the adverse environment created by the acute inflammatory response." This is clearly not the only role Met. There are many papers showing multiple other roles for Met in regeneration (e.g. Webster and Fan 2013 amongst many others). They need to modify this statement and acknowledge the vast literature on this subject.

8. The authors claim that HGF/Met and Cxcl12/Cxcr4 signaling is autocrine (p. 11). However, they have not explicitly tested this by deleting HGF or Cxcl12 in satellite cells. They need to remove this claim in the Discussion and Abstract.

*Reviewer #2:*

Lahmann et al., focused on cytokines which is dramatically increased in early phase of muscle regeneration. Among them, they investigated the roles of HGF and Cxcl12 using conditional KO mice. Intriguingly, the loss of them did not affect the proliferation ability of satellite cells, but functioned to protect satellite cells from cell death induced by TNF-a. in vivo assay system, authors showed the data indicating the influence of loss of HGF and Cxcl12 were rescued by TNF-a neutralizing antibodies. Most of conclusion is supported by the present data. Please respond the following comments.

1. Gene expression pattern of Cxcl12 is similar with that of TNF-a. While, the peak of HGF expression is at 3 dpi, meaning that the peak of HGF/c-Met signaling is not matched with that of TNF-a. Pro-HGF, biologically inactive HGF form, binds to the ECM. The following paper shows that HGF is stored in normal adult skeletal muscle. Is there a possibility that the stored HGF function to suppress the cell death in the early phase of regeneration? While considering this result, please discuss the different expression pattern of HGF and TNF-a.

HGF/SF is present in normal adult skeletal muscle and is capable of activating satellite cells. Tatsumi R, Anderson JE, Nevoret CJ, Halevy O, Allen RE. Dev Biol. 1998 Feb 1;194(1):114-28.

2. In this study, the impact of Met-null on cell death of satellite cells is critical. Authors showed the remarkable increased number of TUNEL+ cells in coCxcr4/Met satellite cell. While, there is no data showing the relevance between Met-null adn apoptosis in vivo. In order to conclude the protective function of Met, reviewer would like to ask authors to present the data.

3. In Figure 4, necrotic cells were also positive for Propidium Iodide. Reviewer recommends to detect apoptotic cells using TUNEL assay.*Reviewer #3:*

The study presented by Lahmann et al., proposes a specific role of Met and Cxcr4 during muscle regeneration that is distinct from the already reported involvement during muscle cells migration. Instead, the authors use genetic mouse models to show that Met and Cxcr4 cooperate in a cell-autonomous manner to protect muscle stem cells against TNFα-induced damage during repair. The manuscript is well written and the experimental approach is overall of high-standards, involving a large number of genetic models. However, a series of control experiments are needed to solidify the results.

Concerns:

I raise two main concerns: 1. For the majority of the experiments, a constitutive Pax7-Cre line is used. In the case of Met flox, tamoxifen-inducible Pax7-CreERT2 line is used but the appropriate controls are missing. 2. The RNA ISH shown in Figure 1D and 2A need more scrutiny. The overlapping pattern between the different probes is alarming and further controls are needed.

Specifically:

– Figure 1

– Figure 1A, B, C The transcript levels of several cytokines are measured in resting and regenerating muscle. It seems that for Figure 1A and B whole muscle extracts were used whereas 1C results (Met expression) are based on isolated muscle stem cells. It would be informative to look at the expression of the other transcripts in isolated MuSCs, and especially Hgf, as it is later suggested to be acting in an MuSC-autocrine fashion (see Figure 1D).

– Figure 1D The authors conclude that "HGF is produced [exclusively] by muscle stem cells and functions in an autocrine manner during repair".This is based on the RNA ISH that shows Hgf expression exclusively and in all Pax7 cells. Some additional experiments are needed to support unequivocally this conclusion. Hybridization protocols can produce artefacts if there are aggregates or other impurities. The similarity between the Pax7 and the HGF pattern is somewhat worrisome. The authors could combine PAX7 IF with Hgf ISH to confirm their observations. In addition, the double ISH could be performed in resting muscle, where Hgf is supposed to be absent or lower. In any case, quantification of the Pax7+/Hgf+ cells is needed.

– Figure 1 E-L The loss of Pax7, specifically during regeneration, is a very interesting phenotype. Some additional information on the proliferation, differentiation and apoptosis kinetics and status of the mutant cells would give important insights into the role of Met in this context.

One general concern is that the analysis of mice with constitutive Cre (Pax7-Cre here) is always risky. It is true that the authors argue that this is not an issue as "Pax7 is first expressed in progenitors that have already reached their targets" and also show that the number of Pax7 cells and the fiber diameter are the same in the resting muscle between control and coMet (Figure 1E-L). This shows, indeed, that in the Pax7-Cre; Met flox/flox mice there is no major MuSC phenotype, yet it does not exclude that the mutant MuSCs, fibres and all other cell types, for all we know, are identical at the molecular level. In fact, as shown in Figure S1J, K the satellite cells in the resting muscle of Pax7-Cre; Met flox/flox mice have half Pax7 transcript and protein.

The authors are well aware of this problem and used two different mouse lines with tamoxifen-inducible Pax7-CreERT2, that nicely recapitulated the loss of MuSCs phenotype (Figure S1B-E). From this figure (Figure S1B-E), however, it becomes evident that all the compound mice were compared to the same control. Instead, each mutant mouse should be compared to its corresponding control, and even more so for the Pax7-CT2-FAN knock-in/knock-out allele that is notoriously impacting muscle regeneration (here, the control should be tamoxifen-treated Pax7-CT2-FAN/+; Met +/+ mice).

In summary, I propose that the authors provide the appropriate comparison using the corresponding controls, and if the results hold true, transfer the conditional Pax7-CreERT2 in the main figure and the constitutive in the supplemental.

– Figure 2

Figure 2A: same comments as for Figure 1D. The overlap in the ISH is worrisome.

Nevertheless, a very interesting Cxcr4/Met synergistic muscle regeneration phenotype is described. It is unfortunate that all this series of experiments is performed with a constitutive Pax7-Cre line.

[Editors' note: further revisions were suggested prior to acceptance, as described below.]

Thank you for resubmitting your article “Met and Cxcr4 signals cooperate to protect muscle stem cells against inflammation-induced damage during regeneration" for consideration by *eLife*. Your revised article has been reviewed by 2 peer reviewers, and the evaluation has been overseen by a Reviewing Editor and Didier Stainier as the Senior Editor. The following individual involved in review of your submission has agreed to reveal their identity: So-ichiro Fukada (Reviewer #2).

The reviewers have discussed their reviews with one another, and the Reviewing Editor has drafted this to help you prepare a revised submission. Overall, we found that this is an extremely responsive revision to the critiques of the previous submission, the manuscript is substantially improved, and the manuscript is an important and original contribution to our understanding of the role of Met and Cxc4 signaling in muscle stem cells during regeneration. Only two small revisions to the manuscript are requested.

Essential Revisions:

1) Figure 7D and S7A: Please, avoid bar plots showing only the average, instead include the individual countings (as you have done for all the other plots).

2) The Abstract needs some changes as some sentences are vague or poorly written.

Abstract

Acute muscle injury is followed by an inflammatory response, tissue repair, and the generation of new muscle fibers by resident muscle stem cells. During regeneration, cytokines and growth factors are produced by the various cell types in the muscle that regulate inflammatory cell behavior as well as muscle stem cell activity, a process well characterized in murine injury models. A better understanding of the function of cytokines and growth factors might be useful to stimulate muscle repair, but needs to distinguish the factor's effects on the different cell types that participate in the repair process [unclear, please re-write]. We show here that MET and CXCR4 cooperate to protect muscle stem cells against TNFα-induced damage during repair. This powerful cyto-protective role was revealed by the genetic ablation of Met and Cxcr4 in muscle stem cells of mice, which resulted in severe apoptosis during the early stages of regeneration. This effect was be [typo] rescued by tnfa neutralizing antibodies. we conclude that muscle stem cells require factors that protect them in the harsh inflammatory environment encountered in acute injury. [too vague for an abstract].

*Reviewer #2:*

In this revised version, one conclusion (autocrine role of Met and Cxcr4 signaling) and the results from gp130-mutatnt mice have been excluded. However, the main conclusion, the protective role of Met/Cxcr4 signaling against TNF-α, has been strengthened. This is an important work in this research field.

*Reviewer #3:*

This is an exemplary revision of a manuscript, where the authors have addressed in depth all the points raised by the 3 reviewers. The revised manuscript includes careful characterization of all possible combinations of genotypes and treatments that serve as robust controls. Also, it is appreciated that the authors have removed the Hgf, following a careful examination of its expression (as suggested by the reviewers). In view of this, the autocrine Met/HGF model is no longer suggested. Despite this modification, the study provides very interesting and original data on a novel, protective role of old players in the muscle field -Met and Cxcr4- that would be of great interest for a broad readership.

Figure 7D and S7A: Please, avoid bar plots showing only the average, instead include the individual countings (as you have done for all the other plots).

---

## [Author Response]

Summary:Lahmann and colleagues examine in "Met and Cxcr4 signals cooperate to protect muscle stem cells against inflammation-induced damage during regeneration" the cell-autonomous role of Met and Cxcr4 signaling in satellite cells during regeneration. The role of Met signaling, in particular, in muscle regeneration has been examined by multiple papers over many years, and roles in activation, proliferation, migration, fusion, and quiescence have all been implicated. The novel finding of this paper is that the authors find that Met and Cxcr4 in satellite cells protects these cells from apoptosis early in regeneration (3 days post injury) in response to TNFα expression. They also suggest that Met and Cxcr4 act cooperatively and HGF/Met and CXCL12/Cxc4 signaling act in an autocrine manner to protect satellite cells. The reviewers were generally enthusiastic about the proposed new role of Met and Cxcr4 in protecting satellite cells from apoptosis during regeneration. The reviewers found that manuscript is well written and the experimental approach is overall of high-standards, involving a large number of genetic models. However, there were several deficiencies noted (particularly with respect to controls) that need to be addressed.

We appreciate the comments about the clarity of the manuscript and about their overall favorable assessment. We have now included a comparison of the phenotypes of the Met mutation (Figures 2 and 3), using as controls the different Pax7Cre lines as requested which is indicated in detail in the point by point response. Moreover, we now used the Pax7iresCreERT2Gaka for all subsequent experiments (Figures 5-7), and the results obtained with this line substantiate and extend our previous findings.

The data supporting that HGF/Met and CXCL12/Cxcr4 functions in an autocrine manner was found to be less well supported and in need of more experimental data to be included in the manuscript.

We were unable to verify the RNAscope data that indicated that Hgf is expressed by activated muscle stem cells using qPCR, and we thank the reviewers for pointing out that these data might be problematic. Neither our qPCR data nor published microarray datasets support the notion that Hgf is expressed at appreciable amounts in activated MuSC. We therefore removed our RNAscope data, and deleted the suggestion that Hgf and Cxcl12 signal in an autocrine manner. The new qPCR data are inserted in Figures 1 and 4 of the revised manuscript.

Essential Revisions:1. A major concern is the issue of the experimental mice used in these experiments. For most experiments Pax7iresCre mice were used to delete Met and/or Cxcr4. As Pax7iresCre deletes in muscle progenitors during development, both the satellite cells and myofibers have Met and Cxcr4 deleted at the start of the regeneration experiments and thus the experiments are not strictly testing the role of Met and/or Cxcr4 only during muscle regeneration. The authors have addressed this in a small number of experiments in Figure 1 and Figure S1. They show that the cross-sectional area of myofibers and the number of satellite cells does not differ between control and Pax7iresCre uninjured muscle, suggesting that loss of Met in Pax7+ cells does not have a major developmental defect. In addition they show (Figure 1K and S1B-E) that in response to injury the number of Pax7+ cells is reduced 7dpi with deletion in Pax7iresCre as well as Pax7iresCreERT2GAKA and Pax7CreERT2FAN mice. Thus they argue that the results with the Pax7iresCre are similar to the results with the tamoxifen-inducible Pax7CreERT2 mice and reflect the role of Met during regeneration and not during development. However, the results in Figure 1K and Figure S1E are not able to be compared because they use different metrics (Pax7+ cells/100 myofibers vs Pax7+ cells/area)

We have now used everywhere in the manuscript the identical metrics (Pax7+ cells/mm2)

and the identity of controls are unclear. Thus whether the results with the Pax7iresCre really reflect the requirement of Met and/or Cxcr4 strictly during regeneration is uncertain. The authors need to repeat key experiments using the Pax7CreERT2FAN or Pax7CreERT2GAKA mice.

As requested by the reviewers, we now included consistently a Cre control in the Figures comparing Met phenotypes observed with different Cre lines (Figures 2, 3). Moreover, we repeated all experiments of the paper using an inducible CreERT2 allele, i.e. Pax7iresCreERT2^GAKA^. Except in Figures 2 and 3 were we compare phenotypes observed with the three Cre alleles, we now entirely rely on Pax7iresCreERT2^GAKA^ for our analysis (Figures 5-7).

To make the manuscript more accessible to readers, we continue to abbreviate the genotypes in Results and Figures. The abbreviations are introduced in Results and Methods, and the exact genotypes are also described in the legend to each Figure. In particular, the following abbreviations are used:

Tx^Gaka^Met mutant: Pax7^iresCreERT2Gaka/+^;Met^flox/flox^ treated with tamoxifen

Tx^Fan^Met mutant: Pax7^CreERT2Fan/+;^Met^flox/flox^ treated with tamoxifen

coMet mutant: Pax7iresCre/+;Met^flox/flox^

Controls for Tx^Gaka^Met: Pax7^iresCreERT2Gaka/+^;Met^+/+^ treated with tamoxifen

Controls for Tx^Fan^Met: Pax7^CreERT2Fan/+^;Met+/+ treated with tamoxifen

Controls for coMet: Pax7^iresCre/+^;Met+/+

Importantly, in our hands Met phenotypes obtained by the use of Pax7iresCreERT2^GAKA^ and the constitutive Pax7iresCre alleles consistently show identical phenotypes. This contrasts the results obtained with Pax7CreERT2^Fan^. The fact that phenotypes observed with Pax7iresCreERT2^Gaka^ and Pax7iresCre are identical indicate to us that these data are reliable.

In Pax7CreERT2^Fan^ allele, the Pax7 coding sequence is disrupted, resulting in lowered Pax7 levels. In contrast, in the Pax7iresCreERT2^Gaka^ and Pax7iresCre alleles, Pax7 remains intact and the CreERT2/Cre sequences, respectively, are fused via an ires sequence which does not affect Pax7 expression. We are not the first to observe that the Pax7CreERT2^FAN^ allele is problematic (reviewer # 3 writes …..”that is notoriously impacting muscle regeneration”; see also von Maltzahn et al., 2013; Mademtzoglou et al. 2018; Noguchi et al., 2019). Although we do not consider this to be a major finding of our work, we wanted to document the impact of the Pax7CreERT2^Fan^ allele on the Met phenotype because we believe that our observations might be of useful for others.

2. There is concern about whether the appropriate control mice have been used in the genetic experiments. Throughout all figure panels, the full genotype of all control and experimental mice should be displayed. Particularly in experiments using Pax7CreERT2FAN, the control mice should be Pax7CreERT2Fan/+; Met+/+ and with tamoxifen. The issue of the control mice used was particularly of concern in Figure S1E.

As the reviewer requested, we include now for all data additional genotypes as controls.

Controls for Tx^Gaka^Met: Pax7^iresCreERT2Gaka/+^;Met+/+ treated with tamoxifen

Controls for Tx^Fan^Met: Pax7^CreERT2Fan/+^;Met+/+ treated with tamoxifen

Controls for coMet: Pax7^iresCre^/+;Met+/+

Importantly, we realized during the revision that many fibers in tamoxifen treated Tx^Fan^Met mice observed after regeneration are ‘ghost fibers’, i.e. represent remnants of laminin matrices devoid of a myosin-positive cell. We therefore use consistently in the revised manuscript pan-myosin and laminin antibodies to quantify fiber diameters, counting only those laminin circles that contain a myosin-positive cell. In the original submission, we had used only laminin antibodies, and thus had not eliminated ghost fibers. The new approach used for quantifications further augmented differences in fiber diameters between Tx^Gaka^Met and coMet mutations on one side, and the Tx^Fan^Met mutation on the other side.

3. The authors conclude that HGF/Met and CXCL12/Cxcr4 signaling is autocrine in satellite cells. However, they do not provide enough data to support such a conclusion. They show in Figures 1D and 2A by smFISH that HGF and CXCl12 is co-expressed with Pax7 in satellite cells, although there was some concern about the quality of this data. The expression of HGF and CXCL12 in satellite cells could be strengthen by examining their expression via qPCR in isolated satellite cells rather than in whole muscle homogenates.

Indeed, we were unable to verify the RNAscope data that indicated that Hgf is expressed by activated muscle stem cells using qPCR, and we thank the reviewers for pointing out that these data might be problematic. Neither our qPCR data nor published microarray datasets support the notion that Hgf is expressed at appreciable amounts in activated MuSC. However, our qPCR data and published microarray datasets support the notion that Cxcl12 is expressed by activated muscle stem cells. The qPCR data are now inserted in Figures 1 and 4 of the revised manuscript, and the RNAscope data were deleted.

Even with such data, the authors can only suggest, but not conclude, that HGF and CXCL12 function in an autocrine manner. HGF and CXCL12 from other cell types may be critical. A definitive test would require that HGF or CXCL12 are conditionally deleted in satellite cells via Pax7CreERT2.

We agree with the reviewer. This was changed in the abstract and other parts of the revised manuscript.

4. The authors propose that Met and Cxcr4 act cooperatively to prevent TNFa-mediated apoptosis. While the authors show the number of apoptotic Pax7+ cells is increased in Pax7iresCre/+; Metfl/fl; Cxcr4fl/fl (Figure 3C), they do not quantify the number of apoptotic Pax7+ cells in Pax7iresCre/+; Metfl/fl or Pax7iresCre/+; Cxcr4fl/fl. Also the control genotype for these experiments is not detailed. The data for all four genotypes needs to be included in order for a role of Met and Cxcr4 cooperativity to be assessed.

We now quantified the number of apoptotic Pax7+ cells in mice of the following genotypes (Figure 6 of the revised manuscript):

Control: Pax7^iresCreERT2Gaka^

Tx^Gaka^Cxcr4: Pax7^iresCreERT2Gaka^/+;Cxcr4^flox/flox^

Tx^Gaka^Met: Pax7^iresCreERT2Gaka/+;^Met^flox/flox^

Tx^Gaka^Cxcr4;Met: Pax7^iresCreERT2Gaka/+^;Cxcr4^flox/flox^;Met^flox/flox^ (all tamoxifen treated).

The data show that the Tx^Gaka^Met mutation impairs survival of stem cells mice, whereas the Tx^Gaka^Cxcr4 mutation does not. Moreover, the double mutation Tx^Gaka^Cxcr4;Met further augments the apoptosis phenotype observed in Tx^Gaka^Met. Thus, the new data provided in the revised manuscript (Figure 6) support and extend the previous data.

5. Figure 4A-C shows an increase in propidium iodide+ satellite cells cultured in the presence of TNFa, which is rescued when either HGF, CXCL12, or HGF and CXCL12 are added. Propidium iodide is an assay for nonviable cells. The authors should conduct this experiment with TUNEL assay (as in Figure 3A-C).

As requested, we replaced the propidium iodide straining and use TUNEL assays in the revised manuscript (Figure 7 A-D); please note that the proportion of dead/apoptotic cells are similar, regardless whether we use propidium iodide or TUNEL.

In addition, these data suggest that either HGF or CXCL12 are sufficient to rescue cell death and there is no additive benefit to using both HGF and CXCL12. This does not support the contention that HGF and CXCL12 are both required to protect satellite cells from TNFα -induced apoptosis.

We show in Figure 7 A-D a rescue of TNF-α-induced apoptosis in a cell culture setting using freshly isolated muscle stem cells kept in low serum on Matrigel; Matrigel is known to contain many growth factors. Thus, the cells are kept under conditions where they are exposed to undefined signals which might not be identical to those present in vivo. Under such conditions, both HGF and Cxcl12 rescue from apoptosis, but there is no additive effect. In contrast to this cell culture experiment, we observe in the in vivo setting a clear additive effect- i.e. stronger apoptosis phenotypes in the conditional Tx^Gaka^Cxcr4;Met double mutant animals than in single Tx^Gaka^Met and TxGakaCxcr4 mutants. Moreover, we show that TNFalpha neutralizing antibodies rescue apoptosis in Tx^Gaka^Met and Tx^Gaka^Cxc4;Met animals. We therefore conclude that in vivo, Met/Cxcr4 act cooperatively. We point out explicitly in the revised manuscript that cooperativity is observed in vivo but not in vitro (page 9, end of 1^st^ and 2nd paragraphs).

6. The in vivo role of Met and Cxcr4 in protection against TNFa -induced apoptosis needs to be strengthened. The authors need to show in vivo whether loss of Met alone leads to an increase in satellite cell apoptosis at 3 dpi. Also, the rescue experiments using neutralizing TNFa antibody (Figure 4D-F) only assay the number of Pax7+ cells/area and not changes in numbers of apoptotic Pax7+ cells; this should be included.

In the revised manuscript, we now provide data using neutralizing antibodies for Tx^GAKA^Met, Tx^Gaka^Cxcr4;Met and control (Pax7^iresCreERT2Gaka^ treated with tamoxifen) animals. This demonstrates that the neutralizing antibodies rescue apoptosis in both, Tx^Gaka^Met and Tx^Gaka^Cxcr4;Met animals. Moreover, we show that the rescue is accompanied by a reduction in the number of apoptotic cells in the muscle of such animals. The new data are shown in Figure 7 of the revised manuscript and extend and support our previous findings.

Reviewer #1:Lahmann and colleagues examine in "Met and Cxcr4 signals cooperate to protect muscle stem cells against inflammation-induced damage during regeneration" the cell-autonomous role of Met and Cxcr4 signaling in satellite cells during regeneration. The role of Met signaling, in particular, in muscle regeneration has been examined by multiple papers over many years, and roles in activation, proliferation, migration, fusion, and quiescence have all been implicated. The novel finding of this paper is the implication that Met and Cxcr4 protects satellite cells from apoptosis early in the regenerative response (3 days post injury) due to TNFa expression. They also suggest that Met and Cxcr4 act cooperatively and in an autocrine manner to protect satellite cells. The finding of a role for Met and Cxcr4 for cooperatively blocking apoptosis during regeneration is interesting, but in need of further data to support. There is little data in this to support that HGF/Met and CXCL12/Cxc4 functions in an autocrine manner. See specific comments.

We thank the reviewer for finding the new role of Met/Cxcr4 in the protection of muscle stem cells against TNF-α induced damage interesting. We included additional data and controls to support these findings.

Moreover, the reviewer is right to point out that the data do not show that Met/Cxcr4 act in an autocrine manner. We had provided data that the ligands/receptors are produced by the same cells, which we were however unable to verify for Hgf by qPCR (Figure 1 of the revised manuscript). We have changed the wording in the abstract/manuscript accordingly.

1. The authors show by qPCR that satellite cells express Met and by smFish that some satellite cells express HGF and conclude "HGF I (probably a typo – "is") produced by muscle stem cells and functions in an autocrine manner during repair" p.5. Based on their expression data and conditional deletion in satellite cells, they can not conclude this. They show that Met is required in satellite cells, but the source of HGF may be from satellite cells or many other cell types present in regenerating muscle that they have not tested.

Indeed, we were unable to verify the RNAscope data that indicated that Hgf is expressed by activated muscle stem cells using qPCR, and we thank the reviewers for pointing out that these data might be problematic. Neither our qPCR data nor published microarray datasets support the notion that Hgf is expressed at appreciable amounts in activated MuSC. As the reviewer requested, we show the qPCR data (Figures1 and 4 of the revised manuscript). Moreover, the RNAscope data were deleted.

2. Figure 1E-L: 1. In these experiments they have deleted Met using the Pax7iresCre, and so Met has been deleted in muscle progenitors throughout development. They need to explicitly state this in the last paragraph of p. 5. I find it surprising that there is no developmental phenotype (neither number of Pax7+ muscle progenitors or myofiber cross-sectional area is affected). Please make sure to highlight this in text.

In the revised manuscript, we mention on page pg. 5 (last paragraph) that the Pax7iresCreinduced Met mutation (coMet) deletes in muscle progenitors throughout development.

Met has a very important migratory function in developing myogenic progenitor cells, that we and others (including the lab of the reviewer) have observed. Pax7 is however not expressed in muscle progenitor cells before or during migration, and is only induced after the progenitors reach their targets (Relaix et al., 2004). Therefore, the Pax7iresCre-induced mutations neither affect the migratory behavior nor the muscle that derive from migrating cells.

Others suggested that Met regulates in addition to migration the size of the precursor pool for secondary myogenesis in the trunk (Maina et al., 1996). For this study, a hypomorph allele that mutates the Grb2 binding site of Met was used. Several possibilities might explain that we do not observe this in our experiments: (1) Allele differences or possibly differences in the genetic background might account for this. We use an allele in which the exon that encodes the ATP-binding site of Met is deleted and that abolishes Met functions. (2) The hypomorph allele that Maina et al. used also affects placental development (page 535 of (Maina et al., 1996): ‘The corresponding placentae appeared slightly smaller compared with controls but were well vascularized (data not shown)’). A small placenta might impair the supply of nutrients/oxygen to the embryos thus indirectly the development of other tissues, including muscle.

Our data presented here are in accordance with earlier findings of our lab. In chimeric mice (chimera of Met+/+ and Met-/- cells), Met-/- cells do not contribute to muscle generated by migrating progenitors (diaphragm, limb muscle) but contribute normally to other muscle groups in the trunk and head (Bladt et al., 1995). Moreover, aggregation chimera of tetraploid (wild type) and diploid (Met−/−) morulae were performed in our lab; tetraploid wildtype cells only contribute to extraembryonic structures including the placenta, whereas the embryo proper is exclusively generated by diploid Met-/- cells. Such chimera lack muscle groups that derive from migratory precursor cells, but display otherwise normal skeletal muscles in the trunk at E17.5 (Dietrich et al., 1999).

Also, please put the actual genotype of control and experimental mice on the Figure panels – do not use the abbreviation "control" and "coMet"; we need to see the actual genotypes of these mice.

I find the exact genotypes in the text and figures difficult. i.e. exact genotypes are long and difficult to understand for non-geneticists. After lengthy discussions, we decided to continue to use abbreviations; the abbreviations are explained in the main text, in Methods, and in the legend of every figure. Thus, the reader can verify genotypes easily, but the flow of the text is not interrupted, and we can label in figures individual panels using a font size that is not too small. In summary, the following abbreviations are used:

Tx^Gaka^Met: Pax7^iresCreERT2Gaka/+;^Met^flox/flox^ treated with tamoxifen

Tx^Fan^Met: Pax7^CreERT2Fan/+^;Met^flox/flox^ treated with tamoxifen

coMet: Pax7^iresCre/+;^Met^flox/flox^

Controls for Tx^Gaka^Met: Pax7^iresCreERT2Gaka/+;^Met^+/+^ treated with tamoxifen

Controls for Tx^Fan^Met: Pax7^CreERT2Fan/+^;Met^+/+^ treated with tamoxifen

Controls for coMet: Pax7^iresCre/+^;Met^+/+^.

The authors state that "similar deficits were observed when Met was mutated in adult muscle stem cells using the tamoxifen-inducible Pax7iresCreERT2GAKA allele". Only the reduction in Pax7+ satellite cells at 7 dpi is shown and not the changes in myofiber cross-sectional area; this should be shown.

In the revised manuscripts, we show number of Pax7+ cells before injury and 7 day after injury, and fiber diameters 7 and 20 dpi for Met mutations induced by the three different Cre alleles (Figures 2 and 3; supplemental data Figure 2 —figure supplement 1). Genotypes used are:

Tx^Gaka^Met: Pax7^iresCreERT2Gaka/+;^Met^flox/flox^ treated with tamoxifen

Tx^Fan^Met: Pax7^CreERT2Fan/+^;Met^flox/flox^ treated with tamoxifen

coMet: Pax7^iresCre/+^; Met^flox/flox^

Controls for Tx^Gaka^Met: Pax7^iresCreERT2Gaka/+^;Met^+/+^ treated with tamoxifen

Controls for Tx^Fan^Met: Pax7^CreERT2Fan/+;^Met^+/+^ treated with tamoxifen

Controls for coMet: Pax7^iresCre/+;^ Met^+/+^.

Importantly, in our hands the data obtained using Pax7iresCreERT2^Gaka^ and Pax7iresCre are always very similar; in Figure 2 and 3 we directly compare the mutations using the three Cre alleles, but otherwise we show in the revised manuscript only data using Pax7iresCreERT2^Gaka^. The constitutive Pax7iresCre-induced mutations were shown in the original manuscript, but are no longer included in Figures5-8 of the revised manuscript to avoid redundancies. The fact that phenotypes observed with Pax7iresCreERT2^Gaka^ and Pax7iresCre are consistently similar indicate to us that these data are reliable.

3. The authors write on p. 6 that "loss of Met in muscle stem cells results in a mild regeneration deficit that is accompanied by a reduction in the number of muscle stem cells". However, the authors never analyze any muscle regeneration phenotypes after 7 dpi. Certainly the work of Webster and Fan 2013 shows a severe regeneration phenotype in the myofibers at later time points (20 dpi). The authors need to look at later time points post injury or explicitly acknowledge the work of Webster and Fan, which clearly shows a regeneration defect.

Webster and Fan show very severe regeneration phenotype at 10 and 20 days after injury in Figure 1 or their paper. As the reviewer requested, we include in the revised manuscript one further time point, 20 dpi, at which we compare the different alleles, in addition to 7dpi time point. At both time points, we observe very severe phenotypes using the Pax7CreERT2^Fan^ allele, and milder phenotypes when the mutation was introduced by Pax7iresCreERT2^Gaka^ or the constitutive Pax7iresCre alleles (Figures 2, 3 and Figure 3 —figure supplement 1, revised manuscript). These data show that we reproduce the results of Webster and Fan when we use the Pax7CreERT2^Fan^ allele to mutate Met, but not when we use Pax7iresCreERT2^Gaka^ or the constitutive Pax7Cre alleles.

Together, our experiments show that the severe phenotype of the Pax7CreERT2^Fan^; Metflox/flox mice is *not only* due to the loss of Met in muscle stem cells, but arises also because of the use of the Pax7CreERT2^Fan^ allele. As the reviewer requested, we acknowledge the work by Webster and Fan on page 6 of the revised manuscript and cite their paper, but also point out that with other Cre lines we cannot detect such a severe phenotype.

4. Figure 2: 1. The case of cooperativity between Met and Cxcr4 would be made easier to see if they included the data on Pax7iresCre;Cxcr4fl/fl mice in the main figure and not in Figure S2. 2. It is unclear if there is increased fibrosis if both Met and Cxcr4 are deleted, versus individual loss of Met and Cxcr4. If they want to make this point they need to include all 4 genotypes (Control that is specified; Pax7iCre; Cxcr4fl/fl; Pax7iCre;Metfl/fl; and Pax7iCre;Cxcr4fl/fl;Metfl/fl) and quantify the degree of fibrosis.

As the reviewer requested, we now show data on the cooperativity in one figure of the revised manuscript to ease comparison of the genotypes (Figure 5-7 and the corresponding figure supplements).

Genotypes displayed are:

Control: Pax^7iresCreERT2Gaka^

Tx^Gaka^Cxcr4: Pax7^iresCreERT2Gaka/+;^Cxcr4^flox/flox^

Tx^Gaka^Met: Pax7^iresCreERT2Gaka/+;^Met^flox/flox^

Tx^Gaka^Cxcr4;Met: Pax7^iresCreERT2Gaka/+^;Cxcr4^flox/flox^;Met^flox/flox^

All genotypes were tamoxifen treated.

Shown are the effects of the mutation on Pax7+ cell numbers, fiber diameter, fibrosis. The new data extend and support our previous conclusion.

5. Figure 3: Panels A-C. The authors show an increase in the number of Tunel+Pax7+ cells at 3dpi in Pax7iCre; Cxcr4fl/fl;Metfl/fl mice. The authors need to show the quantification of Tunel+Pax7+ cells at 3 dpi for Pax7iCre; Cxcr4fl/fl and Pax7iCre;Metfl/fl mice. Also please write the genotype of the "control" mice.

As requested by the reviewer, we now quantified the number of apoptotic Pax7+ cells inmice with the following genotypes:

Control: Pax^7iresCreERT2Gaka/+^

Tx^Gaka^Cxcr4: Pax7^iresCreERT2Gaka/+^;Cxcr4^flox/flox^

Tx^Gaka^Met: Pax7^iresCreERT2Gaka/+^;Met^flox/flox^

Tx^Gaka^Cxcr4;Met: Pax7^iresCreERT2Gaka/+^;Cxcr4^flox/flox^;Met^flox/flox^

All Genotypes were tamoxifen treated.

The new data extend and support our previous conclusion and are shown in Figure 6 of the revised manuscript.

Panels D-H. The authors need to show the number of Pax7+ and BrdU^+^Pax7+ cells for all four genotypes: 1. control, which needs to be specified; 2. Pax7iCre; Cxcr4fl/fl; 3. Pax7iCre;Metfl/fl; and 4. Pax7iCre;Cxcr4fl/fl;Metfl/fl. Without all four genotypes, it is not possible to infer whether the effects of Met and Cxcr4 really are cooperative.

As requested by the reviewer, we now quantified the number of Pax7+ and EdU+Pax7+ cells in mice with different genotypes:

Control: Pax7^iresCreERT2Gaka/+^

Tx^Gaka^Cxcr4: Pax7^iresCreERT2Gaka/+^Cxcr4^flox/flox^

Tx^Gaka^Met: Pax7^iresCreERT2Gaka/+^;Met^flox/flox^

Tx^GakaA^Cxcr4;Met: Pax7^iresCreERT2Gaka/+;^Cxcr4^flox/flox^;Met^flox/flox^

All genotypes were tamoxifen treated.

The new data extend and support our previous conclusion and are displayed in Figure 6 – figure supplement 1.

6. Figure 4: The authors show in Panel C that either Cxcl12 or HGF alone is sufficient to rescue TNFa induced satellite cell death in culture and the effects of Cxcl12 and HGF do not lead to a further rescue. Thus it is most parsimonious to argue that Cxcl12 and HGF do not have a synergistic (cooperative) or additive effect on rescue of cell death – either factor will work.

We show in Figure 7 A-D a rescue of TNF-α-induced apoptosis in a cell culture settingusing freshly isolated muscle stem cells kept in low serum on Matrigel; Matrigel is known to contain many growth factors. Thus, the cells are kept under conditions where they are exposed to undefined signals which might not be identical to those present in vivo. Under such conditions, both HGF and Cxcl12 rescue from apoptosis, but there is no additive effect. In contrast to this cell culture experiment, we observe in the in vivo setting a clear additive effect- i.e. stronger apoptosis phenotypes in the conditional Tx^Gaka^Cxcr4;Met double mutant animals than in single Tx^Gaka^Met and Tx^Gaka^Cxcr4 mutants. Moreover, we show that TNFalpha neutralizing antibodies rescue apoptosis in Tx^Gaka^Met and Tx^Gaka^Cxc4;Met animals. We therefore conclude that in vivo, Met/Cxcr4 act cooperatively. We point out explicitly in the revised manuscript that cooperativity is observed in vivo but not in vitro (page 9, end of 1^st^ and 2nd paragraphs).

Panel D. The authors show that blocking TNFa partially rescues the number of Pax7+ cells when Cxcr4 and Met are deleted in Pax7+ cells. It is important for the authors to show the effects of TNFa blockade on Pax7iCre; Cxcr4fl/fl and Pax7iCre;Metfl/fl, if there is increased apoptosis in these genotypes (see comments above).

As the reviewer requested, we show data on the effects of the TNF-α blocking antibodies on mice of the following genotypes:

Pax7^iresCreERT2Gaka/+^

Pax7^iresERT2CreGaka/+^;Met^flox/flox^ and

Pax7^iresERT2CreGaka/+^;Cxcr4^flox/flox^;Met^flox/flox^

(all tamoxifen treated) (Figure 7).

We have not used it on the Pax7iresCreERT2^Gaka/+;^Cxcr4^flox/flox^ genotypes, because we did not observe increased apoptosis in this genotype. The new data extend and support our previous conclusion.

7. The authors claim that the "major role" of Met and Cxcr4 is to "work together in order to protect stem cells against the adverse environment created by the acute inflammatory response." This is clearly not the only role Met. There are many papers showing multiple other roles for Met in regeneration (e.g. Webster and Fan 2013 amongst many others). They need to modify this statement and acknowledge the vast literature on this subject.

Indeed, there are many papers on the role of Met and Hgf in muscle regeneration- InPubMed I find 2,051 on skeletal muscle AND Met, 264 skeletal muscle AND Hgf, 101 papers on Met AND skeletal muscle AND regeneration, and 102 papers on Hgf AND skeletal muscle AND regeneration (the search was done on May 16th, 2021). We had a short look at all of the ones that deal with regeneration during the revision.

A multitude of these reports describe effects of HGF on cultured myogenic cells from various species (primary muscle cells, myoblasts, myoblast cell lines): Exogenous HGF elicits responses that extend from increased migration, changes in cell morphology, proliferation, cell survival, suppression of differentiation – typical responses to growth factors. It is often difficult to extrapolate from such in vitro work to the roles of a growth factor in vivo. We have cited some of the earliest papers of this literature that describes in vitro effects of HGF on myogenic cells, but I feel strongly that we cannot cite all of the published papers.

Very few of these papers use genetic models to assess muscle regeneration after mutation of Met in muscle stem cells: (1) the Webster/Fan paper (Webster and Fan 2013) on a regenerative phenotype of Met; we cite this paper but provide data that the phenotype of Webster/Fan is not only caused by the Met mutation but also by the use of the Pax7CreERT2^Fan^ allele, and that the severe phenotype cannot be observed when Met is mutated using other Cre alleles. (2) Two papers by the Rando lab on the entry of muscle stem cells into G^alert^ and a role of Met in this process. These papers do not investigate to what extend the Met mutation in stem cells affects regeneration of the muscle. We now cite one of these papers (Rodgers et al., 2014). To further discuss the Met literature on muscle, an extra paragraph was inserted into the revised manuscript (page 10, 2nd paragraph of the discussion).

From our analysis we can conclude that a major function of Met in muscle regeneration is to keep the muscle stem cell numbers ‘normal’ by protecting them from apoptosis, which is extensively documented in our paper. The reduction of the stem cell numbers in the Met mutant (using Pax7iresCreERT2^Gaka^ or the Pax7iresCre) is not caused by an inability to enter the proliferative phase or by a proliferate deficit. We performed an analysis of the proliferation rate (EdU incorporation) of Tx^Gaka^Cxcr4, Tx^Gaka^Met, Tx^Gaka^ACxcr4;Met animals, including the appropriate control. Proliferation rates are slightly increased in Tx^Gaka^Met and Tx^GakaA^Cxcr4;Met mice, possibly due to compensatory proliferation (Figure 6 – figure supplement 1 of the revised manuscript).

We have therefore rephrased the sentence and say more specifically: Unexpectedly, our analysis of the in vivo function of Met and Cxcr4 demonstrates an important cooperative role in muscle repair that protects stem cells against the adverse environment created by the acute inflammatory response (pg. 9 of the revised manuscript). I hope that this satisfies the reviewer.

In regards to additional literature on Met in muscle regeneration we want to point out the following:

Many papers describe sources of HGF during muscle regeneration as well as effects on cell types other than muscle stem cells. As possible sources of circulating factor, different cell types are discussed, as well a release from extracellular matrix. Extracellular matrix as a source of HGF in tissue injury was to my knowledge first proposed in liver regeneration experiments where a very fast increase of HGF is observed that too fast to be due to the synthesis of new transcripts/protein and secretion from cells. This was subsequently again discussed in for many other organs. We cited the first liver papers that introduced the concept in our original manuscript and an early work on HGF in the matrix of the muscle that was pointed out to use by one of the reviewers (Shimomura et al., 1995; Tatsumi et al., 1998, Page 10 of the revised manuscript).

In addition, there are many papers describing (mostly beneficial) roles of exogenous HGF during muscle regeneration in animal models. In these studies, HGF is providedexogenously, either alone or in combinations with other factors (e.g. as serum from patients with liver injuries). Apparently, when purified HGF is used, effects differ depending on the time the factor is given: muscle stem cell numbers were increased, but fiber growth was not enhanced (Miller et al., 2000). We cite this study in the revised version of the manuscript (discussion, page 12). Further studies have investigated HGF in non-myogenic cell types during muscle regeneration, for instance immune cells, endothelia, neurons, mesenchymal stem cells, bone marrow stem cells. We do not cite these reports because we did not investigate roles of HGF/Met in such cell types.

8. The authors claim that HGF/Met and Cxcl12/Cxcr4 signaling is autocrine (p. 11). However, they have not explicitly tested this by deleting HGF or Cxcl12 in satellite cells. They need to remove this claim in the Discussion and Abstract.

Indeed, we were unable to verify the RNAscope data that indicated that Hgf is expressed by activated muscle stem cells using qPCR, and we thank the reviewers for pointing out that these data might be problematic. Neither our qPCR data nor published microarray datasets support the notion that Hgf is expressed at appreciable amounts in activated MuSC. However, our qPCR data and published microarray datasets indicate that Cxcl12 is expressed in muscle stem cells. The qPCR data are now inserted in Figures1 and 4 of the revised manuscript, and the RNAscope data were deleted (Figures 1 and 4 of the revised manuscript).

Reviewer #2:Lahmann et al., focused on cytokines which is dramatically increased in early pahse of muscle regeneration. Among them, they investigated the roles of HGF and Cxcl12 using conditional KO mice. Intriguingly, the loss of them did not affect the proliferation ability of satellite cells, but functioned to protect satellite cells from cell death induced by TNF-a. in vivo assay system, authors showed the data indicating the influence of loss of HGF and Cxcl12 were rescued by TNF-a neutralizing antibodies. Most of conclusion is supported by the present data. Please respond the following comments.1. Gene expression pattern of Cxcl12 is similar with that of TNF-a. While, the peak of HGF expression is at 3 dpi, meaning that the peak of HGF/c-Met signaling is not matched with that of TNF-a. Pro-HGF, biologically inactive HGF form, binds to the ECM. The following paper shows that HGF is stored in normal adult skeletal muscle. Is there a possibility that the stored HGF function to suppress the cell death in the early phase of regeneration? While considering this result, please discuss the different expression pattern of HGF and TNF-a.HGF/SF is present in normal adult skeletal muscle and is capable of activating satellite cells.Tatsumi R, Anderson JE, Nevoret CJ, Halevy O, Allen RE. Dev Biol. 1998 Feb 1;194(1):114-28.

We thank the reviewer for pointing out this study which we cite in the revised paper. We discuss that matrix-bound factor released by proteases might be active during regeneration.

2. In this study, the impact of Met-null on cell death of satellite cells is critical. Authors showed the remarkable increased number of TUNEL+ cells in coCxcr4/Met satellite cell. While, there is no data showing the relevance between Met-null adn apoptosis in vivo. In order to conclude the protective function of Met, reviewer would like to ask authors to present the data.

As the reviewer suggested, we included data on apoptosis in the single mutants, Tx^Gaka^Met and Tx^Gaka^CXCR4, the double mutant Tx^Gaka^Met;CXCR4 and as controls the Pax7CreERT2^Gaka^ in Figure 6 of the revised manuscript. The additional data confirm and extend our previous conclusion.

3. In Figure 4, necrotic cells were also positive for Propidium Iodide. Reviewer recommends to detect apoptotic cells using TUNEL assay.

As suggested by the reviewer, we use TUNEL staining to detect apoptotic cells in vitro in the revised manuscript (Figure 7). Please note that the two assay methods give identical relative values.

Reviewer #3:The study presented by Lahmann et al., proposes a specific role of Met and Cxcr4 during muscle regeneration that is distinct from the already reported involvement during muscle cells migration. Instead, the authors use genetic mouse models to show that Met and Cxcr4 cooperate in a cell-autonomous manner to protect muscle stem cells against TNFα-induced damage during repair. The manuscript is well written and the experimental approach is overall of high-standards, involving a large number of genetic models. However, a series of control experiments are needed to solidify the results.

We thank the reviewer for the overall positive assessment of our paper, and for the praise on the clarity of the manuscript and the quality of the experimental approach.

Concerns:I raise two main concerns: 1. For the majority of the experiments, a constitutive Pax7-Cre line is used. In the case of Met flox, tamoxifen-inducible Pax7-CreERT2 line is used but the appropriate controls are missing.

We acknowledge that the controls used were not described in detail in the original manuscript. In part this was done to keep the manuscript clear- it would be very tedious to mention all controls that were done. In the revised manuscript, we use the following specific genotypes:

For comparison of different cre alleles:

Tx^Gaka^Met: Pax7^iresCreERT2Gaka/+^;Met^flox/flox^ treated with tamoxifen

Tx^Fan^Met: Pax7^CreERT2Fan/+^;Met^flox/flox^ treated with tamoxifen

coMet: Pax7^iresCre/+;^Met^flox/flox^

Controls for Tx^Gaka^Met: Pax7^CreERT2Gaka/+^;Met^+/+^ treated with tamoxifen

Controls for Tx^Fan^Met: Pax7^CreERT2Fan/+^;Met^+/+^ treated with tamoxifen

Controls for coMet: Pax7^iresCre/+^; Met^+/+^.

For the comparison of the effects of met, Cxcr4 and Cxcr4/Met double mutants:

Control: Pax7iresCreERT2^Gaka/+;^

Tx^Gaka^Cxcr4: Pax7iresCreERT2^Gaka/+;^Cxcr4^flox/flox^

Tx^Gaka^Met: Pax7iresCreERT2^Gaka/+;^Met^flox/flox^

Tx^Gaka^Cxcr4;Met: Pax7^iresCreERT2Gaka/+^;Cxcr4^flox/flox^;Met^flox/flox^

Abbreviations used for genotypes are introduced in Results and Methods, and the exact information is also provided in the legend of the figures of the revised manuscript.

2. The RNA ISH shown in Figure 1D and 2A need more scrutiny. The overlapping pattern between the different probes is alarming and further controls are needed.

We were unable to verify the RNAscope data that indicated that Hgf is expressed by activated muscle stem cells using qPCR, and we thank the reviewers for pointing out that these data might be problematic. As requested, we preformed qPCR, and in addition reanalyzed previously published microarray data generated others. Neither our qPCR data nor published microarray datasets support the notion that Hgf is expressed at appreciable amounts in activated MuSC. We therefore removed our RNAscope data, and deleted the suggestion that Hgf and Cxcl12 signal in an autocrine manner. The qPCR data are now inserted in Figures1 and 4, and the ISH data were removed.

Specifically:– Figure 1– Figure 1A, B, C The transcript levels of several cytokines are measured in resting and regenerating muscle. It seems that for Figure 1A and B whole muscle extracts were used whereas 1C results (Met expression) are based on isolated muscle stem cells. It would be informative to look at the expression of the other transcripts in isolated MuSCs, and especially Hgf, as it is later suggested to be acting in an MuSC-autocrine fashion (see Figure 1D).

As suggested by the reviewer, we performed a qPCR experiments (see answer to the previous point of the reviewer).

-Figure 1D The authors conclude that "HGF is produced [exclusively] by muscle stem cells and functions in an autocrine manner during repair".This is based on the RNA ISH that shows Hgf expression exclusively and in all Pax7 cells. Some additional experiments are needed to support unequivocally this conclusion. Hybridization protocols can produce artefacts if there are aggregates or other impurities. The similarity between the Pax7 and the HGF pattern is somewhat worrisome. The authors could combine PAX7 IF with Hgf ISH to confirm their observations. In addition, the double ISH could be performed in resting muscle, where Hgf is supposed to be absent or lower. In any case, quantification of the Pax7+/Hgf+ cells is needed.

See previous comment.

– Figure 1 E-L The loss of Pax7, specifically during regeneration, is a very interesting phenotype. Some additional information on the proliferation, differentiation and apoptosis kinetics and status of the mutant cells would give important insights into the role of Met in this context.

We have included in the revised manuscript data on proliferation, differentiation and apoptosis of Tx^Gaka^Met, Tx^Gaka^Cxcr4 and Tx^Gaka^Cxcr4;Met double mutant at 4 dpi; as a control for this, Pax7iresCreERT2^Gaka/+^; Cxcr4^+/+^;Met^+/+^ mice treated with tamoxifen were used.

We agree with the reviewer that apoptosis kinetics would be interesting to analyze.

However, in early stages of muscle regeneration, antibody stainings do not work well due to high background, presumably due to the fact that immune cells and antibodies are present at high concentrations in the tissue. Thus, a time course of surviving cells cannot be reliably done by combining Pax7 antibodies and TUNEL staining.

A time course of the disappearance of PAX7+ cells could be reliably done only by FACS using a fluorescent marker in muscle stem cells (e.g. Tg:Pax7nGFP, RosafloxstopfloxdtTomato), which would require to cross in an additional allele into all of the strains used (Tx^Gaka^Met, Tx^Gaka^Cxcr4 and Tx^Gaka^Cxcr4;Met), a very time consumable breeding. We have therefore refrained from doing this.

One general concern is that the analysis of mice with constitutive Cre (Pax7-Cre here) is always risky. It is true that the authors argue that this is not an issue as "Pax7 is first expressed in progenitors that have already reached their targets" and also show that the number of Pax7 cells and the fiber diameter are the same in the resting muscle between control and coMet (Figure 1E-L). This shows, indeed, that in the Pax7-Cre; Met flox/flox mice there is no major MuSC phenotype, yet it does not exclude that the mutant MuSCs, fibres and all other cell types, for all we know, are identical at the molecular level. In fact, as shown in Figure S1J, K the satellite cells in the resting muscle of Pax7-Cre; Met flox/flox mice have half Pax7 transcript and protein.The authors are well aware of this problem and used two different mouse lines with tamoxifen-inducible Pax7-CreERT2, that nicely recapitulated the loss of MuSCs phenotype (Figure S1B-E). From this figure (Figure S1B-E), however, it becomes evident that all the compound mice were compared to the same control. Instead, each mutant mouse should be compared to its corresponding control, and even more so for the Pax7-CT2-FAN knock-in/knock-out allele that is notoriously impacting muscle regeneration (here, the control should be tamoxifen-treated Pax7-CT2-FAN/+; Met +/+ mice).In summary, I propose that the authors provide the appropriate comparison using the corresponding controls, and if the results hold true, transfer the conditional Pax7-CreERT2 in the main figure and the constitutive in the supplemental.

In order to address concerns with controls, and to overcome the concerns that arose due to the use of a constitutive Cre line, we have redone many of the experiments, and use now exclusively the following genotypes in the revised manuscript:

For comparison of different cre alleles:

Tx^Gaka^Met: Pax7^iresCreERT2Gaka/+^;Met^flox/flox^ treated with tamoxifen

Tx^Fan^Met: Pax7^CreERT2Fan/+^;Met^flox/flox^ treated with tamoxifen

coMet: Pax7^iresCre/+;^ Met^flox/flox^

Controls for Tx^Gaka^Met: Pax7^CreERT2Gaka/+^;Met^+/+^ treated with tamoxifen

Controls for Tx^Fan^Met: Pax7^CreERT2Fan/+^;Met^+/+^ treated with tamoxifen

Controls for coMet: Pax7^iresCre/+^; Met^+/+^.

For the comparison of the effects of Met, Cxcr4 and Cxcr4/Met double mutants:

Control: Pax7^iresCreERT2Gaka/+^

Tx^Gaka^Met: Pax7^iresCreERT2Gaka/+^;Met^flox/flox^

Tx^Gaka^Cxcr4: Pax7^iresCreERT2Gaka/+^;Cxcr4^flox/flox^

Tx^Gaka^Cxcr4;Met: Pax7^iresCreERT2Gaka/+^;Cxcr4^flox/flox^;Met^flox/flox^

I hope this addresses the concerns of the reviewer.

– Figure 2Figure 2A: same comments as for Figure 1D. The overlap in the ISH is worrisome.

We were unable to verify the RNAscope data that indicated that Hgf is expressed by activated muscle stem cells using qPCR, and we thank the reviewers for pointing out that these data might be problematic. Our qPCR data indicate however that Cxcl12 is expressed by muscle stem cells, which is in accordance to previously published microarray data. The qPCR data are now inserted in Figures1 and 4 of the revised manuscript, and the RNAscope data were deleted.

Nevertheless, a very interesting Cxcr4/Met synergistic muscle regeneration phenotype is described. It is unfortunate that all this series of experiments is performed with a constitutive Pax7-Cre line.

As pointed out above, we have reinvestigated the Met, Cxcr4 and Cxcr4/Met double mutants using the Pax7iresCreERT2^Gaka^ allele (Figures2-7). I hope this addresses the concerns of the reviewer.

References

Bladt F, Riethmacher D, Isenmann S, Aguzzi A, Birchmeier C. 1995. Essential role for the cmet receptor in the migration of myogenic precursor cells into the limb bud. *Nature* 376: 768-771.

Dietrich S, Abou-Rebyeh F, Brohmann H, Bladt F, Sonnenberg-Riethmacher E, Yamaai T, Lumsden A, Brand-Saberi B, Birchmeier C. 1999. The role of SF/HGF and c-Met in the development of skeletal muscle. *Development* 126: 1621-1629.

Heinrich PC, Behrmann I, Muller-Newen G, Schaper F, Graeve L. 1998. Interleukin-6-type cytokine signalling through the gp130/Jak/STAT pathway. *Biochem J* 334 ( Pt 2):297-314.

Mademtzoglou D, Asakura Y, Borok MJ, Alonso-Martin S, Mourikis P, Kodaka Y, Mohan A, Asakura A, Relaix F. 2018. Cellular localization of the cell cycle inhibitor Cdkn1c controls growth arrest of adult skeletal muscle stem cells. *eLife* 7.

Maina F, Casagranda F, Audero E, Simeone A, Comoglio PM, Klein R, Ponzetto C. 1996. Uncoupling of Grb2 from the Met receptor in vivo reveals complex roles in muscledevelopment. *Cell* 87: 531-542.

Noguchi YT, Nakamura M, Hino N, Nogami J, Tsuji S, Sato T, Zhang L, Tsujikawa K, Tanaka T, Izawa K et al. 2019. Cell-autonomous and redundant roles of Hey1 and HeyL in muscle stem cells: HeyL requires Hes1 to bind diverse DNA sites.*Development* 146.

O'Shea JJ, Plenge R. 2012. JAK and STAT signaling molecules in immunoregulation and immune-mediated disease. *Immunity* 36: 542-550.

Relaix F, Rocancourt D, Mansouri A, Buckingham M. 2004. Divergent functions of murine Pax3 and Pax7 in limb muscle development. *Genes Dev* 18: 1088-1105.

Rodgers JT, King KY, Brett JO, Cromie MJ, Charville GW, Maguire KK, Brunson C, Mastey N, Liu L, Tsai CR et al. 2014. mTORC1 controls the adaptive transition of quiescent stem cells from G0 to G(Alert). *Nature* 510: 393-396.

von Maltzahn J, Jones AE, Parks RJ, Rudnicki MA. 2013. Pax7 is critical for the normal function of satellite cells in adult skeletal muscle. *Proc Natl Acad Sci U S A* 110:16474-16479.

Webster MT, Fan CM. 2013. c-MET regulates myoblast motility and myocyte fusion during adult skeletal muscle regeneration. *PLoS One* 8: e81757.

Yu H, Lee H, Herrmann A, Buettner R, Jove R. 2014. Revisiting STAT3 signalling in cancer: new and unexpected biological functions. *Nat Rev Cancer* 14: 736-746.